# Safe Reinforcement Learning using Action Projection: Safeguard the Policy or the Environment?

**Hannah Markgraf**[1,*]    **Shambhuraj Sawant**[3]    **Hanna Krasowski**[2]    **Lukas Schäfer**[1]
**Sébastien Gros**[3]    **Matthias Althoff**[1]

[*] *Corresponding author: hannah.markgraf@tum.de*

[1] *Technical University of Munich,* [2] *University of California, Berkeley,*
[3] *Norwegian University of Science and Technology*

**Reviewed on OpenReview:** *https://openreview.net/forum?id=DDrGSEYxGU*

## Abstract

Projection-based safety filters, which modify unsafe actions by mapping them to the closest safe alternative, are widely used to enforce safety constraints in reinforcement learning (RL). Two integration strategies are commonly considered: Safe environment RL (SE-RL), where the safeguard is treated as part of the environment, and safe policy RL (SP-RL), where it is embedded within the policy through differentiable optimization layers. Despite their practical relevance in safety-critical settings, a formal understanding of their differences is lacking. In this work, we present a theoretical comparison of SE-RL and SP-RL. We identify a key distinction in how each approach is affected by action aliasing, a phenomenon in which multiple unsafe actions are projected to the same safe action, causing information loss in the policy gradients. In SE-RL, this effect is implicitly approximated by the critic, while in SP-RL, it manifests directly as rank-deficient Jacobians during backpropagation through the safeguard. Our contributions are threefold: (i) a unified formalization of SE-RL and SP-RL in the context of actor-critic algorithms, (ii) a theoretical analysis of their respective policy gradient estimates, highlighting the role of action aliasing, and (iii) a comparative study of mitigation strategies, including a novel penalty-based improvement for SP-RL that aligns with established SE-RL practices. Empirical results support our theoretical predictions, showing that action aliasing is more detrimental for SP-RL than for SE-RL. However, with appropriate improvement strategies, SP-RL can match or outperform improved SE-RL across a range of environments. These findings provide actionable insights for choosing and refining projection-based safe RL methods based on task characteristics.

## 1 Introduction

For safety-critical environments, reinforcement learning (RL) policies have to be verified or safeguarded to ensure safety specifications at all times. Provably safe RL through closest-point projection (Krasowski et al., 2023), also often called safety filtering, is a widely used approach that provides safety guarantees during both training and deployment. The projection operation adjusts unsafe actions to the closest safe action by solving an optimization problem. This operation is usually differentiable (Gros et al., 2020), allowing for two different formulations:

(a) Safe environment RL (SE-RL): The projection is treated as part of the unknown environment dynamics, requiring the critic to understand its effect indirectly. Intuitively, the agent learns an unsafe policy that acts in a safeguarded environment (Hunt et al., 2021).

(b) Safe policy RL (SP-RL): The projection is integrated into the policy itself, meaning that gradients are backpropagated through the safeguard. In SP-RL, the objective is to approximate the optimal safeguarded policy for the original unsafe environment (Pham et al., 2018).

Figure 1 illustrates the structural differences between SE-RL and SP-RL. A key advantage of SE-RL is that it leaves the underlying RL algorithm unchanged, preserving any existing theoretical guarantees (Hunt et al., 2021). On the other hand, SP-RL promises a more direct informing of the agent about the impact of the safeguarding by including the sensitivity of the safe action with respect to the unsafe action in the policy gradient estimator. However, SP-RL requires embedding the safeguard within the policy, typically using differentiable optimization layers (Agrawal et al., 2019), and incurs additional computational cost due to sensitivity analysis. These trade-offs lead to a central question: should one safeguard the environment or the policy?

Although empirical comparisons exist (Pham et al., 2018; Kasaura et al., 2023), a theoretical foundation for understanding the trade-offs between SE-RL and SP-RL is still lacking. In this work, we close this gap by developing a formal framework that clarifies their relationship, identifying both equivalences and key differences. As discussed later in section 6.3, a central difference lies in how both approaches are affected by what we refer to as *action aliasing*: In closest-point projection, multiple unsafe actions are mapped to the same safe action, resulting in identical returns. Action aliasing has been well studied in the context of SP-RL. It has been shown to lead to a rank-deficient Jacobian of the safeguard (Gros et al., 2020), also referred to as the zero-gradient problem (Lin et al., 2021). The zero-gradient problem has been associated with degraded performance (Pham et al., 2018; Bhatia et al., 2019) and is commonly addressed through alternative loss functions (Bhatia et al., 2019; Chen et al., 2021) or modified policy update rules (Pham et al., 2018). Similarly, performance issues linked to action aliasing have been reported for SE-RL (Krasowski et al., 2023), and are often mitigated by introducing penalty terms in the reward function that penalize the distance between the original and projected actions (Wabersich & Zeilinger, 2021; Wang, 2022; Markgraf & Althoff, 2023; Bejarano et al., 2025). However, a formal comparison of the action aliasing effect on learning in SE-RL versus SP-RL, and of the differences in improvement strategies, does not yet exist.

Our contributions are as follows:

- We develop a unified formalization of SE-RL and SP-RL in terms of the underlying Markov decision process (MDP), value functions, and policy gradient estimators;

- We prove that the optimal value functions of SE-RL and SP-RL coincide under mild assumptions;

- We formalize the impact of action aliasing on SE-RL, highlighting how it differs from the known zero-gradient problem induced by action aliasing in SP-RL;

- We identify fundamental differences in how action aliasing is mitigated in SE-RL and SP-RL, and propose a novel remedy for SP-RL that aligns more closely with the penalty-based strategy used in SE-RL;

- We perform an empirical comparison of SE-RL and SP-RL, emphasizing the importance of adaptation strategies for handling action aliasing in both deterministic and stochastic policy settings.

The remainder of this study is structured as follows: In section 2, we provide an overview of the related literature. The theoretical background for SE-RL and SP-RL is established in section 3. We then formalize the problem statement in section 4. Sections 5 and 6 provide a formal definition and a comparative analysis of the two approaches, respectively. We discuss mitigation strategies for action aliasing and suggest a new alternative for SP-RL in section 7, before conducting a thorough experimental evaluation in section 8. We discuss our findings in section 9 before concluding in section 10.

## 2 Related Work

Safe RL augments RL algorithms with mechanisms to increase the probability of learning and deploying safe policies, or to ensure hard safety guarantees for the policy (García & Fernández, 2015). Specifically,

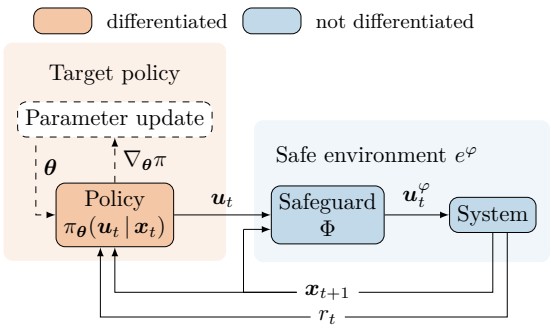 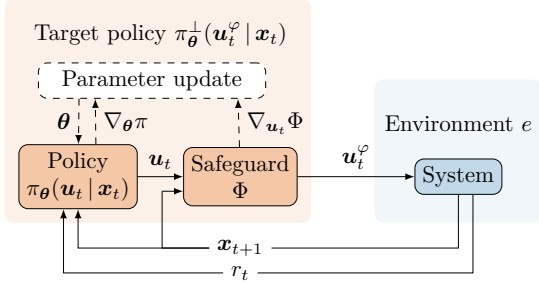

(a) SE-RL: Safeguard is part of the environment.  (b) SP-RL: Safeguard is part of the policy.

Figure 1: In provably safe reinforcement learning, we can either consider the safeguarding as part of the environment (figure 1a) or as part of the policy (figure 1b).

provably safe RL encompasses approaches for which hard safety guarantees are provided during training and deployment (Krasowski et al., 2023). Provably safe RL can be further categorized by means of ensuring that only safe actions are executed. The first category relies on pre-characterizing the set of safe actions through specific structural properties, as it is often necessary to compute center or boundary points efficiently. It comprises methods such as mapping unsafe actions to the interior of the safe action set (Tabas & Zhang, 2022; Kasaura et al., 2023) or sampling from within it (Stolz et al., 2024). The second category of safeguarding mechanisms works directly with the safety constraints themselves, defined, for example, through control barrier functions (Marvi & Kiumarsi, 2022; Wang, 2022) or predictive filters (Selim et al., 2022; Wabersich et al., 2023; Markgraf & Althoff, 2023; Gros et al., 2020). Within this group, closest-point projection is particularly prevalent in continuous control settings, mapping unsafe actions to their nearest safe counterpart by solving a constrained optimization problem.

While effective and widely used, action projection introduces a key limitation that we term action aliasing: All unsafe policy actions that lie within the normal cone to the boundary of the safe action set are projected to the same safe action (see lemma 2). Consequently, they all incur the same reward, no matter how close the policy action was to the safe action. In the context of SE-RL, action aliasing has not been theoretically analyzed, although it has been empirically acknowledged by some studies (Wang, 2022; Krasowski et al., 2023; Markgraf & Althoff, 2023). In contrast, in SP-RL, the impact of action aliasing is well understood. Here, the projection is commonly integrated into the policy using differentiable optimization layers (Agrawal et al., 2019) as the last layer of the policy network to retain gradient flow through the safeguard (Pham et al., 2018; Dalal et al., 2018; Bhatia et al., 2019; Chen et al., 2021; Kasaura et al., 2023). Consequently, action aliasing directly affects the policy gradient computation, eliminating components in the normal direction of the mapping (Gros et al., 2020; Walter et al., 2025).

The approaches for addressing action aliasing differ in SE-RL and SP-RL. A common remedy in SE-RL is to augment the reward with a penalty proportional to the action adjustment (Wabersich & Zeilinger, 2021; Markgraf & Althoff, 2023; Stanojev et al., 2023; Bejarano et al., 2025; Kasaura et al., 2023; Dawood et al., 2025). As shown in Markgraf & Althoff (2023), agents trained with proportional penalties often outperform those trained with constant or no penalties. Recently, Bejarano et al. (2025) confirmed this observation in quadrotor hardware experiments. In SP-RL, additional policy loss terms are often used. For example, Bhatia et al. (2019) employs a loss term that is proportional to the safety constraint violation. Similarly, Chen et al. (2021) proposes a loss term that is proportional to the Euclidean distance between the unsafe and the corresponding safe action. Pham et al. (2018) proposes a two-step gradient step approach in which a first update step is calculated for the unsafe action (with a penalty, following SE-RL), followed by an update step on the safe action.

Most existing studies treat projection safeguarding separately in the context of either SE-RL or SP-RL, leaving a theoretical comparison of the two approaches unexplored. Kasaura et al. (2023) provides a valuable empirical comparison of both basic and improved versions of these methods, focusing on deterministic poli-

cies with static safety constraints. Building on their empirical insights, our work contributes a comprehensive theoretical framework that explains the fundamental differences between SE-RL and SP-RL, including their respective improvement strategies for handling action aliasing. We extend the empirical analysis to include both deterministic and stochastic policies, consider benchmark problems with state-dependent safety constraints, and focus on appropriately scaling penalties in SE-RL and additional loss terms in SP-RL.

## 3 Preliminaries

Policy-based RL algorithms augmented with safety measures have proven successful in safety-critical tasks (Krasowski et al., 2023). Therefore, we provide an overview of the basic concepts of these algorithms before detailing how to differentiate between SE-RL and SP-RL for projection-based safeguarding in section 5.

### 3.1 Reinforcement Learning

An MDP is a tuple $(\mathbb{X}, \mathbb{U}, p_r, p_x, \gamma)$, where $\mathbb{X}$ and $\mathbb{U}$ are the state space and action space. We assume $\mathbf{x}_t$, $\mathbf{u}_t$, and $\mathrm{r}_t$ to be continuous random variables (CRVs) modeling the state, action, and reward at time step $t$, respectively. The transition function $p_x : \mathbb{X} \times \mathbb{U} \times \mathbb{X} \to \mathbb{R}_{\geq 0}$ denotes the probability density $p_x(\boldsymbol{x}_{t+1} \mid \boldsymbol{x}_t, \boldsymbol{u}_t)$ of transitioning to a state $\boldsymbol{x}_{t+1}$ when applying the action $\boldsymbol{u}_t$ in state $\boldsymbol{x}_t$ (Van Hasselt, 2012, section 1.1). The reward for a transition is defined through the probability density $p_r(r_t \mid \boldsymbol{x}_t, \boldsymbol{u}_t)$, where $p_r : \mathbb{X} \times \mathbb{U} \times \mathbb{R} \to \mathbb{R}_{\geq 0}$ (Van Hasselt, 2012, section 1.1). The discount factor $0 \leq \gamma \leq 1$ is used to discount long-term rewards and serves as a simplistic model of the probabilistic lifetime of the MDP.

We define the discounted return at time $t$ as (Sutton & Barto, 2018, equation 3.8)

$$\mathrm{g}_t = \sum_{k=0}^{\infty} \gamma^k \mathrm{r}_{t+k+1}. \tag{1}$$

The goal of RL is to find the policy $\pi$ associated with a probability density function $\pi(\boldsymbol{u}_t \mid \boldsymbol{x}_t)$ over actions given state $\boldsymbol{x}_t$, that maximizes the expected return (Sutton & Barto, 2018, equation 13.4)

$$J(\pi) = \mathbb{E}_\pi \left[ \mathrm{g}_0 \mid \boldsymbol{u}_t \sim \pi(\cdot \mid \boldsymbol{x}_t) \right], \tag{2}$$

where $\mathbb{E}_\pi[\cdot]$ refers to the expected value of a random variable given that the agent follows the policy $\pi$. Note that in the deterministic policy case, states are directly mapped to actions, such that $\boldsymbol{u}_t = \pi(\boldsymbol{x}_t)$.

To find the optimal policy $\pi^* = \arg\max_{\pi \in \Pi} J(\pi)$ (Gros et al., 2020, equation 2), most RL algorithms use various forms of value function estimation. A state value function $v_\pi(\boldsymbol{x}_t) = \mathbb{E}_\pi[\mathrm{g}_t \mid \mathbf{x}_t = \boldsymbol{x}_t]$ (Sutton & Barto, 2018, equation 3.12) expresses how good it is to be in a certain state $\boldsymbol{x}_t$ provided that policy $\pi$ is in use. A state-action value function $q_\pi(\boldsymbol{x}_t, \boldsymbol{u}_t) = \mathbb{E}_\pi[\mathrm{g}_t \mid \mathbf{x}_t = \boldsymbol{x}_t, \mathbf{u}_t = \boldsymbol{u}_t]$ (Sutton & Barto, 2018, equation 3.13) expresses how good it is to use action $\boldsymbol{u}_t$ in state $\boldsymbol{x}_t$, provided that policy $\pi$ is in use after that. Finally, the advantage of taking an action $\boldsymbol{u}_t$ in state $\boldsymbol{x}_t$ is commonly defined as $a_\pi(\boldsymbol{x}_t, \boldsymbol{u}_t) = q_\pi(\boldsymbol{x}_t, \boldsymbol{u}_t) - v_\pi(\boldsymbol{x}_t)$. All optimal policies $\pi^*$ satisfy the same so-called optimal value functions

$$v^*(\boldsymbol{x}_t) = v_{\pi^*}(\boldsymbol{x}_t) \quad \forall \boldsymbol{x}_t \in \mathbb{X}, \tag{3}$$

$$q^*(\boldsymbol{x}_t, \boldsymbol{u}_t) = q_{\pi^*}(\boldsymbol{x}_t, \boldsymbol{u}_t) \quad \forall \boldsymbol{x}_t \in \mathbb{X}, \boldsymbol{u}_t \in \mathbb{U}. \tag{4}$$

### 3.2 Policy Gradient Algorithms

The policy $\pi$ can be inferred directly from the value function, a central concept of so-called value-based RL algorithms. However, these algorithms are mainly designed for discrete action spaces. In continuous and high-dimensional action domains, it is more common to directly learn a policy $\pi_{\boldsymbol{\theta}}$ parameterized by $\boldsymbol{\theta}$, also referred to as policy-based RL. Most modern algorithms use actor-critic algorithms, combining concepts from value-based and policy-based RL by simultaneously learning a policy (the actor) and a value function (the critic) parameterized by $\boldsymbol{\phi}$. To find the optimal policy parameters $\boldsymbol{\theta}^*$, actor-critic algorithms perform repeated updates using $\boldsymbol{\theta} \leftarrow \boldsymbol{\theta} + \alpha \nabla_{\boldsymbol{\theta}} J(\pi_{\boldsymbol{\theta}})$ (Van Hasselt, 2012, equation 13) where $\alpha$ is a learning rate. The

gradient $\nabla_{\boldsymbol{\theta}} J(\pi_{\boldsymbol{\theta}})$ cannot be computed analytically since the reward function and the transition probability distribution are not known explicitly. RL algorithms thus estimate the gradient from data, more specifically from tuples $(\boldsymbol{x}_t, \boldsymbol{u}_t, \boldsymbol{x}_{t+1}, r_t)$. According to the policy gradient theorem, an unbiased estimate of $\nabla_{\boldsymbol{\theta}} J(\pi_{\boldsymbol{\theta}})$ is given by (Silver et al., 2014, theorem 1)

$$\nabla_{\boldsymbol{\theta}} J(\pi_{\boldsymbol{\theta}}) = \mathbb{E}_{\pi} \left[ \nabla_{\boldsymbol{\theta}} \pi_{\boldsymbol{\theta}}(\boldsymbol{x}_t) \nabla_{\boldsymbol{u}_t} q_{\pi,\boldsymbol{\phi}}(\boldsymbol{x}_t, \boldsymbol{u}_t) \right] \tag{5}$$

for deterministic policies and a parameterized critic $q_{\pi,\boldsymbol{\phi}}$, and (Van Hasselt, 2012, equation 15)

$$\nabla_{\boldsymbol{\theta}} J(\pi_{\boldsymbol{\theta}}) = \mathbb{E}_{\pi} \left[ \Psi(\boldsymbol{x}_t, \boldsymbol{u}_t) \nabla_{\boldsymbol{\theta}} \log \pi_{\boldsymbol{\theta}}(\boldsymbol{u}_t \mid \boldsymbol{x}_t) \right] \tag{6}$$

for stochastic policies, where $\Psi$ differs depending on the algorithm. Possible choices include $q_{\pi,\boldsymbol{\phi}}(\boldsymbol{x}_t, \boldsymbol{u}_t); a_{\pi,\boldsymbol{\phi}}(\boldsymbol{x}_t, \boldsymbol{u}_t)$(Schulman et al., 2015), which vary in the variance of the gradient estimate.

Actor-critic algorithms with stochastic policies such as Advantage Actor Critic (A2C) (Mnih et al., 2016) or Proximal Policy Optimization (PPO) (Schulman et al., 2017) that learn a state value $v_{\pi,\boldsymbol{\phi}}$ use $\Psi = \hat{a}(\boldsymbol{x}_t, \boldsymbol{u}_t)$, where $\hat{a}$ is an estimate of the true advantage function. The most common estimator is the generalized advantage estimation (GAE) (Schulman et al., 2015, equation 16)

$$\hat{a}_t^{\text{GAE}} = \sum_{l=0}^{\infty} (\gamma\lambda)^l \delta_{t+l}^v, \tag{7}$$

where $\lambda$ is a hyperparameter and $\delta_t^v$ is the temporal difference residual (Schulman et al., 2015, equation 10)

$$\delta_t^v = r_t + \gamma v_{\pi,\boldsymbol{\phi}}(\boldsymbol{x}_{t+1}) - v_{\pi,\boldsymbol{\phi}}(\boldsymbol{x}_t). \tag{8}$$

To find the optimal parameters $\boldsymbol{\phi}^*$ of the value function $q_{\pi,\boldsymbol{\phi}}$, actor-critic algorithms minimize the expected squared error between the current value function estimate and a target $y$ using (Lillicrap et al., 2015, equation 4)

$$L_{\boldsymbol{\phi}} = \mathbb{E} \left[ \underbrace{(r_t + \gamma q_{\pi,\tilde{\boldsymbol{\phi}}}(\boldsymbol{x}_{t+1}, \boldsymbol{u}_{t+1})}_{y} - q_{\pi,\boldsymbol{\phi}}(\boldsymbol{x}_t, \boldsymbol{u}_t))^2 \right], \tag{9}$$

or the equivalent equation for a state value function $v_{\pi,\boldsymbol{\phi}}$. Here, $q_{\pi,\tilde{\boldsymbol{\phi}}}$ is a target critic that is slowly updated using Polyak averaging. Alternative choices for the target include using the return $\text{g}_t$ directly, or more advanced formulations that address overestimation bias using the minimum value of two target critic networks (Fujimoto et al., 2018).

## 4 Problem Statement

We consider a system with dynamics defined through the transition function $p_x$ that is subject to state-dependent safety constraints of the form

$$g(\boldsymbol{x}_t, \boldsymbol{u}_t) \leq \boldsymbol{0}, \tag{10}$$

which must be satisfied at all times. To enforce these constraints, we define a state-dependent safe action set $\mathbb{U}_{\boldsymbol{x}_t}^{\varphi} \subseteq \mathbb{U}$, such that for any $\boldsymbol{u}_t \in \mathbb{U}_{\boldsymbol{x}_t}^{\varphi}$, there exists a policy that, when applied consecutively from time $t+1$ onward, ensures satisfaction of equation 10 at all times $t' \geq t$. We assume that $\mathbb{U}_{\boldsymbol{x}_t}^{\varphi}$ can be represented as

$$\mathbb{U}_{\boldsymbol{x}_t}^{\varphi} = \{\boldsymbol{u}_t \mid s(\boldsymbol{x}_t, \boldsymbol{u}_t) \leq \boldsymbol{0}\}, \tag{11}$$

where constraints $s$ could, for example, be formulated using control barrier functions, predictive filters, or, as in our experiments, robust control invariant (RCI) sets (see appendix A.1). Furthermore, we define $\tilde{\mathbb{X}} \subseteq \mathbb{X}$ as the set of states for which $\mathbb{U}_{\boldsymbol{x}_t}^{\varphi}$ is non-empty.

Given the safe action set, we introduce a safeguard $\Phi : \tilde{\mathbb{X}} \times \mathbb{U} \to \mathbb{U}^{\varphi}$ into the interaction between the RL policy and the system that ensures that unsafe inputs are projected to the constraint boundary by solving

$$\Phi(\boldsymbol{x}_t, \boldsymbol{u}_t) = \operatorname*{arg\,min}_{\tilde{\boldsymbol{u}}_t \in \mathbb{U}} \quad \frac{1}{2} \|\tilde{\boldsymbol{u}}_t - \boldsymbol{u}_t\|_2^2 \tag{12a}$$

$$\text{s.t.} \quad s(\boldsymbol{x}_t, \tilde{\boldsymbol{u}}_t) \leq 0, \tag{12b}$$

where $\boldsymbol{u}_t \sim \pi(\cdot \,|\, \boldsymbol{x}_t)$. As a result, only safe actions $\boldsymbol{u}_t^\varphi = \Phi(\boldsymbol{x}_t, \boldsymbol{u}_t)$ can be applied to the system, and the return in equation 1 depends on a sequence of safe actions and states. Note that the mapping in equation 12 only modifies unsafe actions. To ensure that $\Phi$ is a deterministic operator, we assume that $\mathbb{U}_{\boldsymbol{x}_t}^\varphi$ is convex. This is a mild assumption since safe sets are commonly constructed as convex inner approximations in practice due to their simple parametrization (e.g., polytopes, ellipsoids), computational tractability, and the uniqueness of closest-point projections.

As shown in figure 1, we can use two different architectures to integrate $\Phi$ into the RL training loop to maximize the expected return in equation 2: We can consider the safeguard to be part of the black-box environment dynamics and aim to approximate the optimal unsafe policy $\pi^* \in \Pi$ for the safeguarded environment. Or we can learn a safe policy $\pi^\perp \in \Pi^\perp$, where $\Pi^\perp \subseteq \Pi$ is the set of safe policies. Note that for any safe policy, the probability density function $\pi^\perp(\boldsymbol{u}_t^\varphi \,|\, \boldsymbol{x}_t)$ is zero for all unsafe actions. In this work, we investigate whether one solution approach dominates the other, both theoretically and empirically.

**Notation:**  In the remainder of this work, we will omit the subscript $t$ when possible and use $\boldsymbol{x}, \boldsymbol{x}', \boldsymbol{u}, r$ instead of $\boldsymbol{x}_t, \boldsymbol{x}_{t+1}, \boldsymbol{u}_t, r_t$ for brevity. Furthermore, we will omit the subscripts $\boldsymbol{\theta}$ and $\boldsymbol{\phi}$ to improve readability.

## 5 Perspectives on Safe Reinforcement Learning

We develop a unified theoretical framework for both approaches to safe RL with action projection by specifying the corresponding value functions and policy gradient estimates. An overview is provided in table 1. For brevity, we limit our analysis to RL algorithms derived directly from the policy gradient theorem, as including methods with additional modifications (e.g., PPO, Soft-Actor Critic (SAC)) would require extensive theoretical derivations beyond the scope of this paper. Similar to the work in Gros et al. (2020), we distinguish between stochastic and deterministic policies. We then discuss the equivalences and differences of both approaches in section 6.

### 5.1 Safe Environment Reinforcement Learning (SE-RL)

In SE-RL, the safeguard is considered part of the system dynamics that are unknown to the RL agent. This changes the MDP from section 3.1 to $M^{\text{SE}} = (\tilde{\mathbb{X}}, \mathbb{U}, p_x^{\text{SE}}, p_r^{\text{SE}}, \gamma)$, with $p_x^{\text{SE}} : \tilde{\mathbb{X}} \times \mathbb{U} \times \tilde{\mathbb{X}} \to \mathbb{R}_{\geq 0}$ and $p_r^{\text{SE}} : \tilde{\mathbb{X}} \times \mathbb{U} \times \mathbb{R} \to \mathbb{R}_{\geq 0}$. Please note that the support of these densities changes compared to the original MDP introduced in section 3.1. Here, $p_x^{\text{SE}}(\boldsymbol{x}' \,|\, \boldsymbol{x}, \Phi(\boldsymbol{x}, \boldsymbol{u}))$ and $p_r^{\text{SE}}(r \,|\, \boldsymbol{x}, \Phi(\boldsymbol{x}, \boldsymbol{u}))$ are defined as the composition of the safeguard mapping $\Phi$ with the original transition functions introduced in section 3.1. Accordingly, the environment receives a (potentially) unsafe action $\boldsymbol{u} \sim \pi$ that is projected internally. Consequently, the value functions are learned for the unsafe policy such that

$$q_\pi^{\text{SE}}(\boldsymbol{x}, \boldsymbol{u}) = \mathbb{E}_\pi \left[ \mathrm{g}_t \,|\, \mathbf{x}_t = \boldsymbol{x}, \mathbf{u}_t = \boldsymbol{u} \right], \tag{13a}$$

$$v_\pi^{\text{SE}}(\boldsymbol{x}) = \mathbb{E}_\pi \left[ \mathrm{g}_t \,|\, \mathbf{x}_t = \boldsymbol{x} \right]. \tag{13b}$$

The policy gradient estimate is computed as given in equation 5 and equation 6. Note that the critic forming this estimate is not aware of the safeguard and has to assess its impact through the data alone.

### 5.2 Safe Policy Reinforcement Learning (SP-RL)

If we consider the safeguard to be a part of the policy, sampled actions are adjusted by a differentiable optimization layer, such that the final policy action is a random variable $\mathbf{u}_t^\varphi := \Phi(\mathbf{x}_t, \mathbf{u}_t)$. The transition function and the probability density of the reward of the MDP $M^{\text{SP}}$ for SP-RL are then defined as $p_x^{\text{SP}} : \tilde{\mathbb{X}} \times \mathbb{P}(\mathbb{U}^\varphi) \times \tilde{\mathbb{X}} \to \mathbb{R}_{\geq 0}$ and $p_r^{\text{SP}} : \tilde{\mathbb{X}} \times \mathbb{P}(\mathbb{U}^\varphi) \times \mathbb{R} \to \mathbb{R}_{\geq 0}$, respectively, where the action space $\mathbb{P}(\mathbb{U}^\varphi)$ is the power set of all $\mathbb{U}_{\boldsymbol{x}}^\varphi$. We aim to learn a value function for the safe actions such that

$$q_{\pi^\perp}^{\text{SP}}(\boldsymbol{x}, \boldsymbol{u}^\varphi) = \mathbb{E}_{\pi^\perp} \left[ \mathrm{g}_t \,|\, \mathbf{x}_t = \boldsymbol{x}, \mathbf{u}_t^\varphi = \boldsymbol{u}^\varphi \right], \tag{14a}$$

$$v_{\pi^\perp}^{\text{SP}}(\boldsymbol{x}) = \mathbb{E}_{\pi^\perp} \left[ \mathrm{g}_t \,|\, \mathbf{x}_t = \boldsymbol{x} \right]. \tag{14b}$$

Table 1: Overview of the MDP formulations and learning equations for SE-RL and SP-RL with stochastic and deterministic policies, respectively.

| | SE-RL | SP-RL |
|---|---|---|
| MDP | $M^{\mathrm{SE}} = (\tilde{\mathbb{X}}, \mathbb{U}, p_r^{\mathrm{SE}}, p_x^{\mathrm{SE}}, \gamma)$ 
 $p_x^{\mathrm{SE}} : \tilde{\mathbb{X}} \times \mathbb{U} \times \tilde{\mathbb{X}} \to \mathbb{R}_{\geq 0}$ 
 $p_r^{\mathrm{SE}} : \tilde{\mathbb{X}} \times \mathbb{U} \times \mathbb{R} \to \mathbb{R}_{\geq 0}$ | $M^{\mathrm{SP}} = (\tilde{\mathbb{X}}, \mathbb{P}(\mathbb{U}^\varphi), p_r^{\mathrm{SP}}, p_x^{\mathrm{SP}}, \gamma)$ 
 $p_x^{\mathrm{SP}} : \tilde{\mathbb{X}} \times \mathbb{P}(\mathbb{U}^\varphi) \times \tilde{\mathbb{X}} \to \mathbb{R}_{\geq 0}$ 
 $p_r^{\mathrm{SP}} : \tilde{\mathbb{X}} \times \mathbb{P}(\mathbb{U}^\varphi) \times \mathbb{R} \to \mathbb{R}_{\geq 0}$ |
| Policy | $\pi : \tilde{\mathbb{X}} \times \mathbb{U} \to \mathbb{R}_{\geq 0}$; $\pi : \tilde{\mathbb{X}} \to \mathbb{U}$ | $\pi^\perp : \tilde{\mathbb{X}} \times \mathbb{P}(\mathbb{U}^\varphi) \to \mathbb{R}_{\geq 0}$; $\pi^\perp : \tilde{\mathbb{X}} \to \mathbb{P}(\mathbb{U}^\varphi)$ |
| Value functions | $q_\pi^{\mathrm{SE}}(\boldsymbol{x}, \boldsymbol{u}) = \mathbb{E}_\pi [g_t \,|\, \mathbf{x}_t = \boldsymbol{x}, \mathbf{u}_t = \boldsymbol{u}]$ 
 $v_\pi^{\mathrm{SE}}(\boldsymbol{x}) = \mathbb{E}_\pi [g_t \,|\, \mathbf{x}_t = \boldsymbol{x}]$ | $q_{\pi^\perp}^{\mathrm{SP}}(\boldsymbol{x}, \boldsymbol{u}^\varphi) = \mathbb{E}_{\pi^\perp} [g_t \,|\, \mathbf{x}_t = \boldsymbol{x}, \mathbf{u}_t^\varphi = \boldsymbol{u}^\varphi]$ 
 $v_{\pi^\perp}^{\mathrm{SP}}(\boldsymbol{x}) = \mathbb{E}_{\pi^\perp} [g_t \,|\, \mathbf{x}_t = \boldsymbol{x}]$ |
| Policy gradient estimates | $\nabla_{\boldsymbol{\theta}} J(\pi) = \mathbb{E}_\pi \left[ \Psi_\pi^{\mathrm{SE}}(\boldsymbol{x}, \boldsymbol{u}) \nabla_{\boldsymbol{\theta}} \log \pi(\boldsymbol{u} \,|\, \boldsymbol{x}) \right]$ 
 $\nabla_{\boldsymbol{\theta}} J(\pi) = \mathbb{E}_\pi \left[ \nabla_{\boldsymbol{\theta}} \pi(\boldsymbol{x}) \nabla_{\boldsymbol{u}} q_\pi^{\mathrm{SE}}(\boldsymbol{x}, \boldsymbol{u}) \right]$ | $\nabla_{\boldsymbol{\theta}} J(\pi^\perp) = \mathbb{E}_{\pi^\perp} \left[ \Psi_{\pi^\perp}^{\mathrm{SP}}(\boldsymbol{x}, \boldsymbol{u}^\varphi) \nabla_{\boldsymbol{\theta}} \log \pi(\boldsymbol{u} \,|\, \boldsymbol{x}) \right]$ 
 $\nabla_{\boldsymbol{\theta}} J(\pi^\perp) = \mathbb{E}_{\pi^\perp} \left[ \nabla_{\boldsymbol{\theta}} \pi^\perp(\boldsymbol{x}) \nabla_{\boldsymbol{u}^\varphi} q_{\pi^\perp}^{\mathrm{SP}}(\boldsymbol{x}, \boldsymbol{u}^\varphi) \right]$ |

Note that this requires us to also project the action $\boldsymbol{u}'$ when computing the state-action value $q_{\pi, \tilde{\Phi}}(\boldsymbol{x}', \boldsymbol{u}^{\varphi'})$ for the target $y$ given in equation 9. To obtain the policy gradient estimates, we must distinguish between deterministic and stochastic policies.

### 5.2.1 Deterministic Policies

The deterministic policy gradient estimate in SP-RL is defined by

$$\nabla_{\boldsymbol{\theta}} J(\pi^\perp) = \mathbb{E}_{\pi^\perp} \left[ \nabla_{\boldsymbol{\theta}} \pi^\perp(\boldsymbol{x}) \nabla_{\boldsymbol{u}^\varphi} q_{\pi^\perp}^{\mathrm{SP}}(\boldsymbol{x}, \boldsymbol{u}^\varphi) \right]. \tag{15}$$

Computing the gradient $\nabla_{\boldsymbol{\theta}} J(\pi^\perp)$ requires differentiating through the safeguard $\Phi$. We can apply the chain rule to obtain

$$\nabla_{\boldsymbol{\theta}} \pi^\perp(\boldsymbol{x}) = \nabla_{\boldsymbol{u}} \Phi(\boldsymbol{x}, \boldsymbol{u}) \nabla_{\boldsymbol{\theta}} \pi(\boldsymbol{x}). \tag{16}$$

The sensitivity of the safeguard $\nabla_{\boldsymbol{u}} \Phi(\boldsymbol{x}, \boldsymbol{u})$ can be obtained using the implicit function theorem as described in appendix A.2.

### 5.2.2 Stochastic Policies

To find the optimal parameters of the safeguarded policy $\pi^\perp$ in the stochastic policy case, we use the policy gradient estimate

$$\nabla_{\boldsymbol{\theta}} J(\pi^\perp) = \mathbb{E}_{\pi^\perp} \left[ \Psi_{\pi^\perp}^{\mathrm{SP}}(\boldsymbol{x}, \boldsymbol{u}^\varphi) \nabla_{\boldsymbol{\theta}} \log \pi^\perp(\boldsymbol{u}^\varphi \,|\, \boldsymbol{x}) \right], \tag{17}$$

which depends on the probabilities of safe actions given states as well as on the value function estimate for the safe policy $\Psi_{\pi^\perp}^{\mathrm{SP}}$. While the unsafe policy $\pi(\boldsymbol{u} \,|\, \boldsymbol{x})$ features a bounded probability density, the safe policy $\pi^\perp(\boldsymbol{u}^\varphi \,|\, \boldsymbol{x})$ takes on a Dirac-like structure on the boundary of the safe set as illustrated in Gros et al. (2020, figure 2). In general, no closed-form expression of $\pi^\perp(\boldsymbol{u}^\varphi \,|\, \boldsymbol{x})$ exists, making it difficult to provide a gradient estimate for a projected stochastic policy. However, in Gros et al. (2020, Proposition 2), the authors show that we obtain an unbiased gradient estimate using

$$\nabla_{\boldsymbol{\theta}} J(\pi^\perp) = \mathbb{E}_{\pi^\perp} \left[ \Psi_{\pi^\perp}^{\mathrm{SP}}(\boldsymbol{x}, \boldsymbol{u}^\varphi) \nabla_{\boldsymbol{\theta}} \log \pi(\boldsymbol{u} \,|\, \boldsymbol{x}) \right].[1] \tag{18}$$

---

[1] An alternative for SP-RL with stochastic policies is proposed by Chen et al. (2021). Given an unsafe policy represented by a Gaussian distribution $\mathcal{N}(\boldsymbol{\mu_\theta}, \boldsymbol{\Sigma_\theta})$, they first project the mean using $\boldsymbol{\mu_\theta^\perp}(\boldsymbol{x}) = \Phi(\boldsymbol{\mu_\theta}, \boldsymbol{x})$ and then sample $\boldsymbol{u}^s \sim \mathcal{N}(\boldsymbol{\mu_\theta^\perp}, \boldsymbol{\Sigma_\theta})$. However, since this action is not guaranteed to satisfy the constraints, they project again to obtain $\boldsymbol{u}^\varphi = \Phi(\boldsymbol{u}^s, \boldsymbol{x})$. Since this approach does not solve the problem of finding a closed-form solution for $\pi^\perp(\boldsymbol{u}^\varphi \,|\, \boldsymbol{x})$, it is not further considered here.

# 6 Comparative Analysis of SE-RL and SP-RL

Having established SE-RL and SP-RL as two distinct solution paradigms for the same task, we now examine their theoretical relationship and practical differences. Since the learning equations differ between approaches, identical initial policy and critic parameters will yield different updates, raising a fundamental question: do both frameworks converge to equivalent (sub-)optimal solutions?

## 6.1 Theoretical Equivalence of Optimal Solutions

We first establish that, in principle, both approaches target the same optimal value function.

**Theorem 1.** *For any given task, let $v_\pi^{SE}$ be a value function for policy $\pi$ interacting with the MDP $M^{SE}$ and $v_{\pi^\perp}^{SP}$ a value function for policy $\pi^\perp$ interacting with $M^{SP}$. Then, a value function $v^{SE^*}$ that is optimal for $M^{SE}$ is also an optimal value function $v^{SP^*}$ for $M^{SP}$.*

The proof is provided in appendix A.3. Theorem 1 states that, given a specific task, a policy $\pi^*$ that is optimal in the SE-RL framework yields the same expected return as an optimal policy $\pi^{\perp^*}$ in the SP-RL framework. While this theoretical result provides important insight, it offers limited practical guidance for continuous state and action spaces where optimal policies are rarely found (Van Hasselt, 2012). Subsequently, we derive a more practical result for a certain group of algorithms.

## 6.2 Practical Equivalence for Stochastic Policies

Algorithms employing stochastic policies and GAE (e.g., A2C) learn a state value function $v_\pi(\boldsymbol{x})$ to compute $\hat{a}^{GAE}$. The following lemma shows that in this case, the differences between SE-RL and SP-RL disappear.

**Lemma 1.** *Let $\pi_{\boldsymbol{\theta}}, \pi_{\boldsymbol{\theta}}^\perp$ and $v_{\pi,\boldsymbol{\phi}}^{SE}, v_{\pi^\perp,\boldsymbol{\phi}}^{SP}$ denote the parameterized policies and value functions for a given task when employing SE-RL or SP-RL, respectively. Define the advantage estimates used in the policy gradient estimates in SE-RL (equation 6) and SP-RL (equation 18) as*

$$\Psi_\pi^{SE}(\boldsymbol{x}, \boldsymbol{u}) = \hat{a}^{GAE}(\boldsymbol{x}, \boldsymbol{u}) \quad and \quad \Psi_{\pi^\perp}^{SP}(\boldsymbol{x}, \boldsymbol{u}^\varphi) = \hat{a}^{GAE}(\boldsymbol{x}, \boldsymbol{u}^\varphi),$$

*respectively, where $\hat{a}^{GAE}$ is computed using equation 7. Then, for any initial parameters $\boldsymbol{\theta}_0$ and $\boldsymbol{\phi}_0$, the parameter updates $\boldsymbol{\theta}_{k+1} \leftarrow \boldsymbol{\theta}_k + \alpha\nabla_{\boldsymbol{\theta}} J(\boldsymbol{\theta}_k)$ and $\boldsymbol{\phi}_{k+1} \leftarrow \boldsymbol{\phi}_k + \alpha \nabla_{\boldsymbol{\phi}} L(\boldsymbol{\phi}_k)$ are identical in both the SE-RL and SP-RL framework at every iteration $k \geq 0$.*

The proof is provided in appendix A.4.

**Corollary 1.** *Under the conditions of Lemma 1, the sequences of parameters $(\boldsymbol{\theta}_k, \boldsymbol{\phi}_k)_{k=0}^\infty$ generated by SE-RL and SP-RL are identical, and consequently both frameworks converge to the same policy and value function parameters.*

In contrast, for algorithms using deterministic policies, SE-RL and SP-RL have different learning equations and may thus converge to different locally optimal policies as shown next.

## 6.3 Differences for Deterministic Policies

As summarized in table 1, the first difference consists in the scope of the critic. Most algorithms using deterministic policies learn a state-action value function $q$. In SP-RL, $q$ is evaluated only on safe actions and, thus, does not learn a meaningful approximation outside of the safe action set. In SE-RL, $q$ is also evaluated on unsafe actions. However, any value function in SE-RL that is conditioned on actions is affected by what we refer to as action aliasing:

**Lemma 2** (Action aliasing). *Let $\partial\mathbb{U}_{\boldsymbol{x}}^\varphi$ be the boundary of a convex safe action set $\mathbb{U}_{\boldsymbol{x}}^\varphi$ and $N_{\mathbb{U}_{\boldsymbol{x}}^\varphi}(\boldsymbol{u}^b) = \{\boldsymbol{n} \,|\, \boldsymbol{n}^T(\boldsymbol{u}^\varphi - \boldsymbol{u}^b) \leq 0 \quad \forall \boldsymbol{u}^\varphi \in \mathbb{U}_{\boldsymbol{x}}^\varphi\}$ the normal cone at $\boldsymbol{u}^b \in \partial\mathbb{U}_{\boldsymbol{x}}^\varphi$ (Boyd & Vandenberghe, 2004). Furthermore, let $\mathbb{U}^e(\boldsymbol{u}^b) = \{\boldsymbol{u}^e \in \mathbb{U} \setminus \mathbb{U}_{\boldsymbol{x}}^\varphi \,|\, \boldsymbol{u}^e = \boldsymbol{u}^b + \zeta\boldsymbol{n}, \zeta > 0, \boldsymbol{n} \in N_{\mathbb{U}_{\boldsymbol{x}}^\varphi}(\boldsymbol{u}^b)\}$ be the set of unsafe actions $\boldsymbol{u}^e$ in the normal cone of a given $\boldsymbol{u}^b \in \partial\mathbb{U}_{\boldsymbol{x}}^\varphi$. Then, any action $\boldsymbol{u}^e \in \mathbb{U}^e(\boldsymbol{u}^b)$ will be projected to $\boldsymbol{u}^b$, causing a transition to the same next state $\boldsymbol{x}'$ and yielding the same reward $r^b \sim p_r(r \,|\, \boldsymbol{x}, \boldsymbol{u}^b)$.*

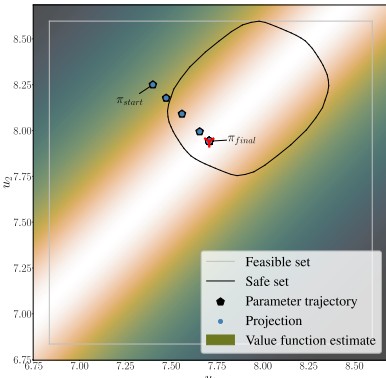 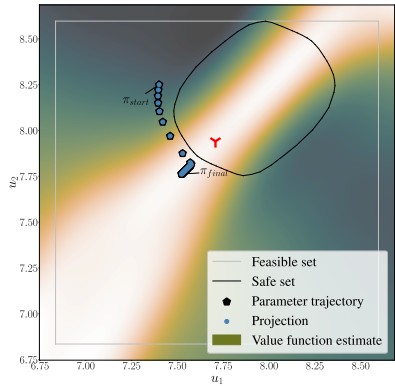 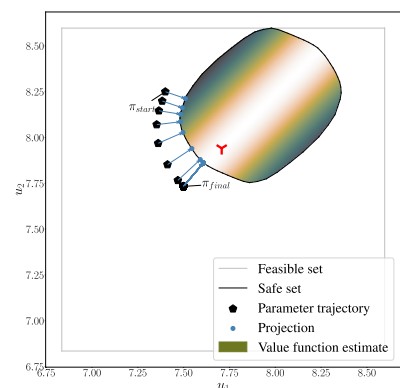

(a) Without safeguarding, the critic approximates the true objective function, and the policy converges to the optimal safe action.

(b) SE-RL: Policy converges to an unsafe and suboptimal action that lies in the direction normal to the safe optimal action and the safe set boundary.

(c) SP-RL: Policy with differentiable projection safeguard does not improve in the direction normal to the projection and therefore does not reach the optimal safe action.

Figure 2: Effect of action aliasing on SE-RL and SP-RL algorithms using deterministic policies. We illustrate the policy improvement step for a given state $\boldsymbol{x}$ in the quadrotor balancing task. The deterministic policy $\pi(\boldsymbol{x})$ is updated for 50 steps using a loss function based on the learned state-action value function $q(\boldsymbol{x}, \boldsymbol{u})$ in the SE-RL case and $q(\boldsymbol{x}, \boldsymbol{u}^\varphi)$ in the SP-RL case.

This means that the critic cannot distinguish between the actions $\boldsymbol{u}^e \in \mathbb{U}^e(\boldsymbol{u}^b)$, potentially hindering learning. We formalize this *flat-lining critic* phenomenon in the following lemma, which is proven in appendix A.5:

**Lemma 3** (Flat-lining critic). *Following lemma 2, the state-action value function, the advantage function, and the GAE adhere to*

$$q_\pi(\boldsymbol{x}, \boldsymbol{u}^e) = q_\pi(\boldsymbol{x}, \boldsymbol{u}^b) \quad \forall \boldsymbol{u}^e \in \mathbb{U}^e,$$

$$a_\pi(\boldsymbol{x}, \boldsymbol{u}^e) = a_\pi(\boldsymbol{x}, \boldsymbol{u}^b) \quad \forall \boldsymbol{u}^e \in \mathbb{U}^e, \text{ and}$$

$$\hat{a}_\pi^{GAE}(\boldsymbol{x}, \boldsymbol{u}^e) = \hat{a}_\pi^{GAE}(\boldsymbol{x}, \boldsymbol{u}^b) \quad \forall \boldsymbol{u}^e \in \mathbb{U}^e,$$

*respectively.*

Note that the flat-lining critic issue also applies to algorithms using stochastic policies in SE-RL. Essentially, lemma 3 states that gradient information along the normal directions will be eliminated by using a perfect value function approximation in equation 5.

We illustrate this using a simplified example. Consider the quadrotor task described in section A.8. Here, we assume deterministic system dynamics and compute the safe action set $\mathbb{U}_{\boldsymbol{x}}^\varphi$ for a fixed state $\boldsymbol{x}$ given the RCI set. We use supervised learning and a random behavior policy to train a critic network to approximate the state-action value function $q_\pi(\boldsymbol{x}, \boldsymbol{u})$ for the given state. Then, we use a deterministic target policy represented by a tensor of the same dimension as $\boldsymbol{u}$ and perform 50 policy update steps using stochastic gradient descent to maximize the learned objective. Note that for this environment, task performance (reaching the equilibrium state) and safety are well-aligned since the equilibrium is considered a safe state. Therefore, the optimal action is often also safe, as shown in figure 2. In appendix A.6, we provide a second example where task performance and safety are not aligned.

Figure 2a shows a non-safeguarded setup, where the critic has learned a good approximation of the true objective function. The policy improvement steps lead to the optimal safe action. However, such a learning process is only possible in computer simulation or non-safety-critical environments, as unsafe actions might be executed. In SE-RL, the safeguarding takes place in the environment such that the projection is not visible in figure 2b. We observe that the policy actions never converge to the safe action set, and would

not reach the optimal safe action even after the projection. This is caused by the flat-lining critic, which eliminates gradients in the direction normal to the boundary of the safe action set.

SP-RL avoids the flat-lining critic by restricting evaluation to safe actions. We visualize this in figure 2c. Instead, the impact of action aliasing is shifted to another part of the policy gradient estimation. When computing the sensitivity of the safeguard $\nabla_{\boldsymbol{u}}\Phi(\boldsymbol{x}, \boldsymbol{u})$, any components in the normal direction to the boundary vanish (Gros et al., 2020), leading to a rank-deficient Jacobian. This is termed the *zero-gradient problem* in existing work (Lin et al., 2021; Kasaura et al., 2023). As shown in figure 6c, the zero-gradient problem can be particularly challenging for safe action sets with non-smooth surfaces, as multiple constraints are active on vertices, further reducing the rank of the Jacobian. This is also visible to a lesser extent for the flat-lining critic in figure 6b.

We would like to highlight that while the zero-gradient problem in SP-RL always exists, the flat-lining critic problem in SE-RL depends on the quality of the learned value function. An imperfect value function approximation may, in fact, alleviate the issue, as shown in other works highlighting the advantages of sampling-based gradient estimation (Suh et al., 2022). We will examine the practical implications in section 8.

## 7 Addressing Action Aliasing in SE-RL and SP-RL

We extend our comparative analysis to variants of SE-RL and SP-RL that aim to address the shared issue of action aliasing. First, we establish how existing approaches differ, impairing the theoretical equivalence established in theorem 1. Then, we propose an alternative approach for SP-RL that re-establishes the validity.

### 7.1 SE-RL

In SE-RL, action aliasing is commonly addressed by adding a penalty $h$ to the reward each time the safeguard has to intervene (Wabersich & Zeilinger, 2021; Markgraf & Althoff, 2023; Stanojev et al., 2023; Bejarano et al., 2025; Kasaura et al., 2023; Dawood et al., 2025). This changes the MDP reward function from section 5.1 to $r^{\mathrm{SE,aug}} : \tilde{\mathbb{X}} \times \mathbb{U} \times \mathbb{R} \to \mathbb{R}_{\geq 0}$ where

$$r^{\mathrm{SE,aug}}(\boldsymbol{x}, \boldsymbol{u}) = r^{\mathrm{SE}}(\boldsymbol{x}, \boldsymbol{u}) - h. \tag{19}$$

A common choice for the penalty function is the squared Euclidean distance between the safe and the unsafe action

$$h = \xi(\boldsymbol{u}^{\varphi}, \boldsymbol{u}) = \begin{cases} 0 & \text{if } \boldsymbol{u} \in \mathbb{U}_{\boldsymbol{x}}^{\varphi}, \\ w\|\boldsymbol{u} - \boldsymbol{u}^{\varphi}\|_2^2 & \text{otherwise,} \end{cases} \tag{20}$$

where $w$ is a hyperparameter. Introducing a penalty changes the optimization goal of SE-RL as shown in figure 3a. To enable a better comparison with the improvement strategies for SP-RL presented in section 7.2, we quantify the impact the penalty has on the value function and policy gradient estimate. The state-action value function $q_\pi^{\mathrm{aug}}(\boldsymbol{x}, \boldsymbol{u})$ can be rewritten as

$$
\begin{aligned}
q_\pi^{\mathrm{aug}}(\boldsymbol{x}, \boldsymbol{u}) &= \mathbb{E}_\pi\left[\sum_{k=0}^{\infty} \gamma^k \mathrm{r}_{t+k+1}^{\mathrm{SE,aug}} \,\middle|\, \mathbf{x}_t = \boldsymbol{x}, \mathbf{u}_t = \boldsymbol{u}\right] \\
&= \mathbb{E}_\pi\left[\sum_{k=0}^{\infty} \gamma^k (\mathrm{r}_{t+k+1}^{\mathrm{SE}} - h_{t+k+1}) \,\middle|\, \mathbf{x}_t = \boldsymbol{x}, \mathbf{u}_t = \boldsymbol{u}\right] \\
&= \underbrace{\mathbb{E}_\pi\left[\sum_{k=0}^{\infty} \gamma^k \mathrm{r}_{t+k+1}^{\mathrm{SE}} \,\middle|\, \mathbf{x}_t = \boldsymbol{x}, \mathbf{u}_t = \boldsymbol{u}\right]}_{q_\pi^{\mathrm{SE}}(\boldsymbol{x}, \boldsymbol{u})} - \underbrace{\mathbb{E}_\pi\left[\sum_{k=0}^{\infty} \gamma^k h_{t+k+1} \,\middle|\, \mathbf{x}_t = \boldsymbol{x}, \mathbf{u}_t = \boldsymbol{u}\right]}_{q_\pi^{\mathrm{pen}}(\boldsymbol{x}, \boldsymbol{u})}.
\end{aligned}
$$

Then, the policy gradient estimate for a deterministic policy can be expressed as

$$\nabla_{\boldsymbol{\theta}} J(\pi) = \mathbb{E}_\pi\left[\nabla_{\boldsymbol{\theta}} \pi(\boldsymbol{x}) \nabla_{\boldsymbol{u}} q_\pi^{\mathrm{aug}}(\boldsymbol{x}, \boldsymbol{u})\right] = \mathbb{E}_\pi\left[\nabla_{\boldsymbol{\theta}} \pi(\boldsymbol{x}) \nabla_{\boldsymbol{u}} \left(q_\pi^{\mathrm{SE}}(\boldsymbol{x}, \boldsymbol{u}) - q_\pi^{\mathrm{pen}}(\boldsymbol{x}, \boldsymbol{u})\right)\right], \tag{21}$$

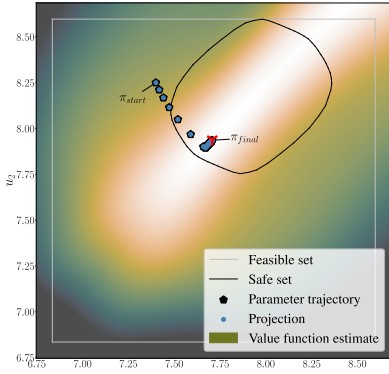 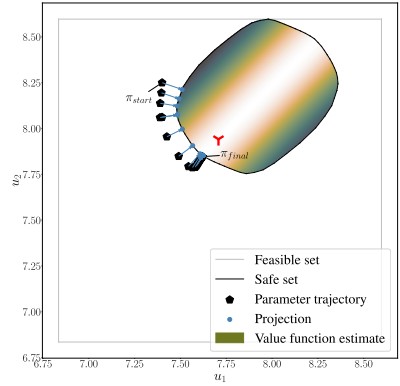 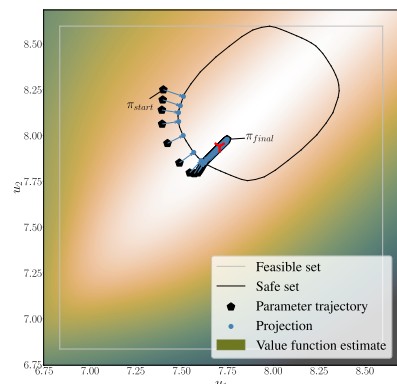

(a) SE-RL: Adding a penalty to the reward addresses the flat-lining critic problem and improves convergence toward the optimal safe action.

(b) SP-RL: An additional policy loss term that penalizes the distance between unsafe and safe action improves convergence to the optimal safe action but stops on the boundary of the safe action set.

(c) SP-RL: Learning an additional penalty critic is very similar to adding a penalty to the reward in SE-RL. Note that as the penalty critic is conditioned on unsafe actions, we display the objective over the entire action space.

Figure 3: Effect of improvement strategies when using a differentiable safeguard during policy updates for a given state $\boldsymbol{x}$ in the navigation task. The deterministic policy $\pi_{\boldsymbol{\theta}}(\boldsymbol{x})$ is updated for 50 steps using a loss function based on the learned state-action value function $q(\boldsymbol{x}, \boldsymbol{u})$ in the SE-RL case and $q(\boldsymbol{x}, \boldsymbol{u}^{\varphi})$ in the SP-RL case.

.

showing the impact of the penalty. The term $q_{\pi}^{\mathrm{pen}}(\boldsymbol{x}, \boldsymbol{u})$ steers the policy away from actions that frequently trigger the safeguard, such that policy actions that are inherently safe or closer to the safe region are preferred. The strength of this effect is controlled by the hyperparameter $w$ in equation 20. A similar derivation can be done for stochastic policies, resulting in

$$\nabla_{\boldsymbol{\theta}} J(\pi) = \mathbb{E}_{\pi} \left[ \Psi_{\pi}^{\mathrm{aug}}(\boldsymbol{x}, \boldsymbol{u}) \nabla_{\boldsymbol{\theta}} \log \pi(\boldsymbol{u} \,|\, \boldsymbol{x}) \right] = \mathbb{E}_{\pi} \left[ (\Psi_{\pi}^{\mathrm{SE}}(\boldsymbol{x}, \boldsymbol{u}) - \Psi_{\pi}^{\mathrm{pen}}(\boldsymbol{x}, \boldsymbol{u})) \nabla_{\boldsymbol{\theta}} \log \pi(\boldsymbol{u} \,|\, \boldsymbol{x}) \right]. \tag{22}$$

## 7.2 SP-RL

In SP-RL, action aliasing manifests as rank-deficient Jacobians of the safeguard that directly enter the policy gradient computation through backpropagation. Unlike in SE-RL, where penalties can influence the value function approximation to mitigate action aliasing effects, reward penalties in SP-RL cannot eliminate the underlying rank-deficiency in the sensitivity of the safeguard. Instead, existing literature in SP-RL suggests a direct modification of the policy loss to improve performance (Chen et al., 2021; Bhatia et al., 2019). For deterministic policies, this results in a combined policy gradient estimate

$$\nabla_{\boldsymbol{\theta}} J(\pi^{\perp}) = \mathbb{E}_{\pi^{\perp}} \left[ \nabla_{\boldsymbol{\theta}} \pi^{\perp}(\boldsymbol{x}) \nabla_{\boldsymbol{u}^{\varphi}} q_{\pi^{\perp}}^{\mathrm{SP}}(\boldsymbol{x}, \boldsymbol{u}^{\varphi}) - \nabla_{\boldsymbol{\theta}} d(\cdot) \right], \tag{23}$$

where $d(\cdot)$ commonly depends on the unsafe and the safe action. For example, in Chen et al. (2021), the authors suggest using the squared distance between the unsafe and the projected action as an additional loss such that $d = \xi(\pi^{\perp}(\boldsymbol{x}), \pi(\boldsymbol{x}))$ as defined in equation 20. Others use a loss term proportional to the constraint violation, which is theoretically similar (Bhatia et al., 2019). We illustrate the effect of the squared distance loss in figure 3b for a deterministic policy. Compared to figure 2c, the policy evolves toward the boundary of the safe action set, even though it does not reach the optimal safe action. For stochastic policies, the vanishing gradient problem does not apply as discussed in section 6.3. Nevertheless, we can apply this additional loss to the mean $\boldsymbol{\mu}$ of the Gaussian policy $\mathcal{N}(\boldsymbol{\mu}, \boldsymbol{\Sigma})$ and its projection $\boldsymbol{\mu}^{\perp}$ such that

$$\nabla_{\boldsymbol{\theta}} J(\pi^{\perp}) = \mathbb{E}_{\pi^{\perp}} [\Psi_{\pi^{\perp}}(\boldsymbol{x}, \boldsymbol{u}^{\varphi}) \nabla_{\boldsymbol{\theta}} \log \pi(\boldsymbol{u} \,|\, \boldsymbol{x})] - \mathbb{E}_{\pi^{\perp}} [\nabla_{\boldsymbol{\theta}} \xi(\boldsymbol{\mu}^{\perp}(\boldsymbol{x}), \boldsymbol{\mu}(\boldsymbol{x}))]. \tag{24}$$

Using an additional loss term as in equation 23 and equation 24 captures only myopic effects of unsafe actions. In contrast, penalties added to the reward are embedded into the value function estimate, providing information regarding their long-term consequences. Consequently, the per-sample loss in SP-RL targets a different optimal value function than the improved SE-RL version presented in section 7.1, breaking the equivalence established in theorem 1. Furthermore, the per-sample loss can only push unsafe actions to the boundary of the safe action set, but not to its interior.

Therefore, we suggest an alternative that consists of training an additional critic conditioned on the unsafe policy such that

$$q_\pi^{\text{pen}}(\boldsymbol{x}, \boldsymbol{u}) = \mathbb{E}_\pi \left[ \text{g}_t^{\text{pen}} \mid \mathbf{x}_t = \boldsymbol{x}, \mathbf{u}_t = \boldsymbol{u} \right], \tag{25}$$

where $\text{g}_t^{\text{pen}} = \sum_{k=0}^\infty \gamma^k h_{t+k+1}$. This yields the deterministic policy gradient estimate

$$\nabla_{\boldsymbol{\theta}} J(\pi^\perp) = \mathbb{E}_{\pi^\perp} \left[ \nabla_{\boldsymbol{\theta}} \pi^\perp(x) \nabla_{\boldsymbol{u}^\varphi} q_{\pi^\perp}^{\text{SP}}(\boldsymbol{x}, \boldsymbol{u}^\varphi) \right] - \mathbb{E}_\pi \left[ \nabla_{\boldsymbol{\theta}} \pi(\boldsymbol{x}) \nabla_{\boldsymbol{u}} q_\pi^{\text{pen}}(\boldsymbol{x}, \boldsymbol{u}) \right], \tag{26}$$

which is similar in structure to the one from equation 21. The main difference to SE-RL is that the first term in equation 26 depends on the safeguarded policy and involves the sensitivity of the safeguard. Since the second term is computed for the unsafe policy, convergence behavior compared to the vanilla loss function is improved as shown in figure 3c. In comparison to the per-sample loss, the policy reaches the interior of the safe action set.

In principle, this approach can also be applied to algorithms with stochastic policies, resulting in the policy gradient estimate

$$\nabla_{\boldsymbol{\theta}} J(\pi^\perp) = \mathbb{E}_{\pi^\perp} \left[ \Psi_{\pi^\perp}(\boldsymbol{x}, \boldsymbol{u}^\varphi) \nabla_{\boldsymbol{\theta}} \log \pi(\boldsymbol{u} \mid \boldsymbol{x}) \right] - \mathbb{E}_\pi \left[ \Psi_\pi^{\text{pen}}(\boldsymbol{x}, \boldsymbol{u}) \nabla_{\boldsymbol{\theta}} \log \pi(\boldsymbol{u} \mid \boldsymbol{x}) \right]. \tag{27}$$

However, as discussed in section 5.2.2, the distinction to SE-RL with additional penalties vanishes when the algorithm is based on learning a state value function $v_{\pi^\perp}^{\text{SE}}(\boldsymbol{x})$ that does not depend on the action. More details on the penalty critic implementation can be found in appendix A.7.

# 8 Experiments

We design our numerical experiments[2] to answer the following questions: (1) Without any modifications, which approach achieves better returns empirically - SE-RL or SP-RL? (2) How do the approaches for addressing action aliasing in SE-RL and SP-RL compare to each other? For SP-RL, we consider the per-sample squared distance loss (PSL) and our proposed penalty critic (PenC), and for SE-RL proportional penalties.

We restrict our investigation to actor-critic algorithms based on the policy gradient theorem, as Gros et al. (2020) provide a derivation of unbiased policy gradient estimates under the SP-RL framework for these algorithms. We choose Twin-delayed Deep Deterministic Policy Gradient (TD3) (Fujimoto et al., 2018) and A2C (Mnih et al., 2016) as RL algorithms using deterministic and stochastic policies, respectively.

We use cvxpylayers (Agrawal et al., 2019) to integrate the safeguard into the policy for SP-RL as it automates the computation of the sensitivities of the projection in the backward pass. To ensure a fair comparison, we run all experiments on the same CPU (Intel Core i9-14900K). Furthermore, we conduct hyperparameter tuning for the unsafe baselines of each algorithm and then use this set of hyperparameters for all experiments. The weighting factor $w$ is an important hyperparameter. Therefore, we do not tune it, but instead test for different choices: $w \in \{0.1, 0.5, 1.0, 2.0\}$. We report the final performance using the interquartile mean and 95% bootstrapped confidence intervals of the undiscounted return over 7 training runs evaluated on 10 random seeds, respectively.

## 8.1 Benchmark Problems

We perform the evaluation using a stabilization task for two classic control examples – a pendulum and a quadrotor – as well as a navigation task, and an energy management system (EMS) optimization task. For

---

[2]Our implementation is available at: https://github.com/TUMcps/serl-sprl.

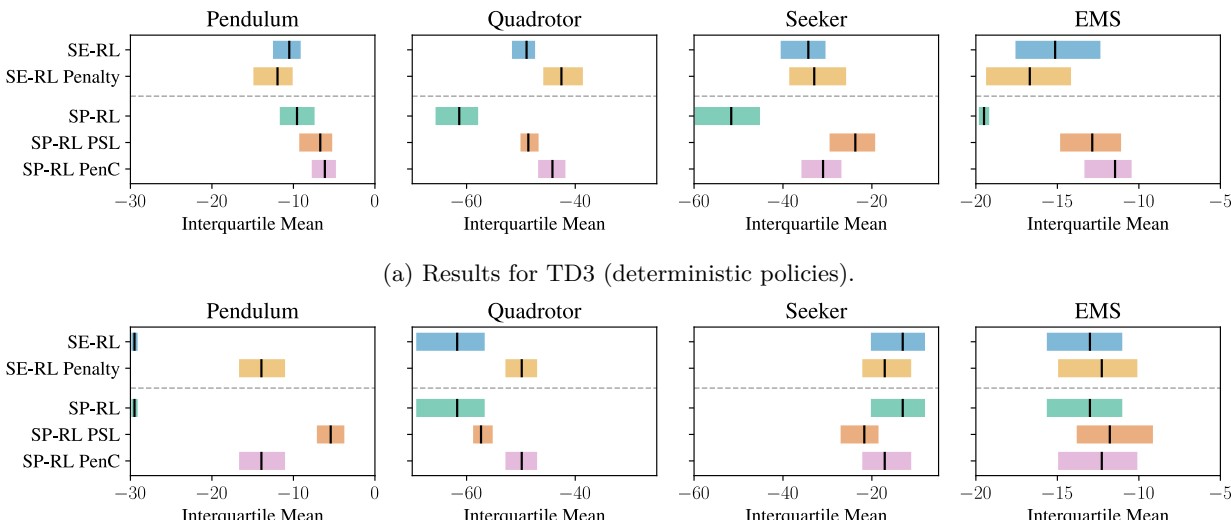

(a) Results for TD3 (deterministic policies).

(b) Results for A2C (stochastic policies). For the pendulum, we clip the returns obtained with vanilla A2C to enable a better comparison. The actual returns are much lower as shown in table 7.

Figure 4: Comparison of vanilla and modified SE-RL/SP-RL approaches (PSL: per-sample loss, PenC: penalty critic). We provide the interquartile mean and the 95% confidence interval of the return achieved at test time. For the improved SE-RL/SP-RL approaches, we only show the result for the best-performing choice of $w \in \{0.1, 0.5, 1.0, 2.0\}$. The full results are listed in appendix A.10.

the first two tasks, we compute RCI sets using the approach from Schäfer et al. (2024). Since the RCI sets are centered around the equilibrium point, task performance is closely aligned with safety. This is different in the navigation task, where a simple point mass seeker has to find the shortest path to a goal while avoiding obstacles, such that the optimal safe policy operates in close proximity to the unsafe regions. Here, instead of an RCI set, we use a state-dependent safe action set to avoid collisions. In the EMS task, there are two competing objectives: maintaining the indoor temperature close to a set point and minimizing electricity cost. Consequently, the alignment of safety and performance depends on which objective dominates at a given point in time. Further details on the benchmark problems are provided in appendix A.8.

## 8.2 Results

### 8.2.1 Comparison of SE-RL and SP-RL Without Modifications

As shown in lemma 1, for algorithms such as A2C that employ stochastic policies and learn a state value function, SE-RL and SP-RL result in the same policies. Consequently, they deliver the same return at test time as shown in figure 4b.

For TD3, figure 4a shows that vanilla SE-RL often outperforms SP-RL, especially for more complex environments. For the pendulum task, the final performance is very similar, with a slight advantage for SP-RL. For the quadrotor, the EMS, and the seeker task, the differences are more pronounced, and vanilla SE-RL clearly dominates SP-RL. Figure 5 shows a large variance in the training performance for TD3-SP-RL caused by several non-convergent runs, which explains the poor performance at test time.

### 8.2.2 Comparison of Improved SE-RL and SP-RL

In figure 4, we only report the results for the best-performing choice of $w$, while the full results are shown in appendix A.10. For the quadrotor and the seeker environment, the choices of $w$ that perform well are largely consistent for one algorithm. For the pendulum, larger $w$ in SE-RL can deteriorate performance significantly, while SP-RL with the per-sample loss is very robust.

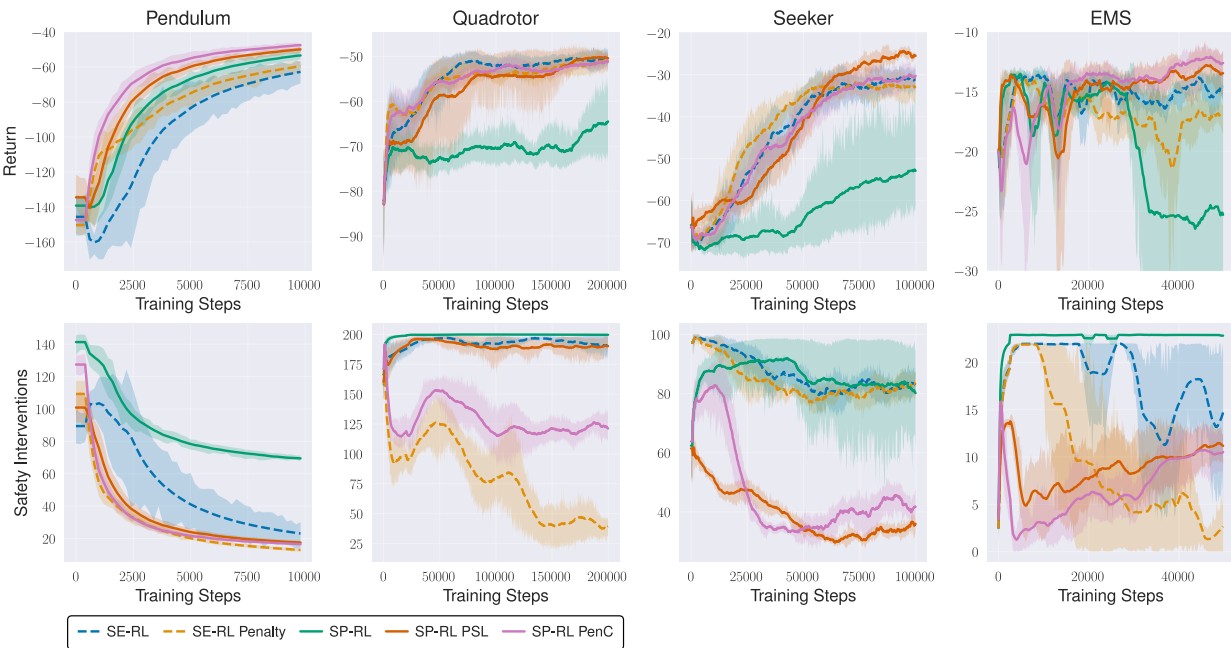

Figure 5: Interquartile mean and 95% bootstrap confidence interval of the return and safeguard interventions over environment steps during training with TD3. We compare the vanilla SE-RL and SP-RL versions and the strategies for mitigating action aliasing.

For TD3-SP-RL, both the per-sample loss and the penalty critic improve test performance across all four environments (see figure 4a). As shown in figure 5, the convergence issues of vanilla TD3-SP-RL in the quadrotor, the EMS, and the seeker can be mitigated. Which of the two strategies for action aliasing performs better depends on the type of environment: In the two balancing tasks and the EMS, the penalty critic delivers better results, while the per-sample loss has the edge in the navigation task. Statistical analysis using Kruskal-Wallis and Dunn's tests with Bonferroni correction ($p < 0.05$) confirmed that compared to TD3-SE-RL with penalties, one of the improved SP-RL versions always performs on par or better.

For A2C, the penalty in SE-RL and the penalty critic in SP-RL are again equivalent. They significantly improve performance for the pendulum and quadrotor environments, and slightly for the EMS. In the seeker environment, performance slightly deteriorates for all improvement strategies in both SE-RL and SP-RL.

## 9  Discussion

For actor-critic algorithms using stochastic policies and GAE, both theoretical and empirical results confirm that SE-RL and SP-RL are equivalent. While the per-sample loss on the policy mean can sometimes improve performance, it is usually outperformed by penalty-based approaches. The seeker environment proved more challenging for improvement strategies due to the difficulty of weighing task performance with safety and the already strong performance of vanilla A2C.

As reported previously (Pham et al., 2018), we observe convergence issues when using SP-RL with deterministic policies, particularly in complex environments, while SE-RL shows no such issues. This supports our theoretical finding that critic flat-lining only hinders learning under perfect value function approximations. In contrast, rank-deficient Jacobians always affect policy updates in SP-RL, making action aliasing more detrimental. Following Bhatia et al. (2019); Chen et al. (2021), we find that these convergence issues can be mitigated by adding loss terms proportional to the distance between safe and unsafe actions.

The relative effectiveness of our proposed penalty critic versus the per-sample loss depends on the alignment between safety and task performance. In the balancing tasks and the EMS, where these objectives align, the

penalty critic outperforms the per-sample loss because it can converge to optimal actions within the safe set, while the per-sample loss is limited to the boundary (figures 3b and 3c). In the seeker environment, where optimal task actions typically lie outside the safe set, both methods converge to boundary actions, making the per-sample loss more efficient.

Overall, improved SP-RL variants typically match or exceed SE-RL performance for deterministic policies, but require 3–12 times the computation time of SE-RL due to sensitivity computations via cvxpylayers. Our wall clock time analysis in appendix A.9 shows that the increase in computation time scales with the complexity of the projection problem (12). We observe the smallest difference between SE-RL and SP-RL for the seeker task, where the constraints (12b) reflect a point containment problem given an unsafe action and a safe action set. This is much simpler than the set containment problem that has to be solved for the pendulum, the EMS, and the quadrotor task (see appendix A.1). Since the state and action space dimensionality of the quadrotor and the EMS are higher than that of the pendulum, they feature the highest gap between SE-RL and SP-RL wall clock times. While for such environments the performance gains in SP-RL have to be weighed against computational complexity, we expect recent advances in the field of differentiable optimization layers (Nguyen & Donti; Grontas et al., 2026; Frey et al., 2025) to mitigate this issue in the future. Furthermore, it is often possible to decouple high-dimensional safety-critical dynamics into several subproblems, which can then be solved in parallel during both the forward and the backward pass (Chen et al., 2021).

## 10 Conclusion

We present a comprehensive theoretical and empirical comparison of SE-RL and SP-RL, two prominent approaches for integrating projection-based safeguards into actor-critic RL. Our unified formalization enables us to prove that both approaches share optimal value functions but differ in their learning dynamics. For a specific subclass of RL algorithms that use stochastic policies and GAE, we show the theoretical and practical equivalence of SE-RL and SP-RL. For algorithms using deterministic policies, the approaches diverge depending on whether the sensitivity of the safeguard mapping is used explicitly in the backward pass of the policy loss (SP-RL), or whether it is learned implicitly through the value function approximation (SE-RL). A central concept when analyzing these differences is action aliasing, where multiple unsafe actions are mapped to identical safe actions. Our analysis reveals that the effect of action aliasing is more detrimental to SP-RL than SE-RL, potentially leading to convergence issues. We propose a novel penalty critic for SP-RL that estimates discounted cumulative penalties proportional to the distance between safe and unsafe actions, providing a principled mitigation strategy aligned with penalty-based approaches in SE-RL. Our theoretical findings and empirical analysis provide practitioners with clear guidance: Vanilla SE-RL presents a strong baseline, particularly for environments where scaling additional penalties or loss terms is challenging, while improved SP-RL variants should be considered when performance gains justify the additional computational cost.

**Acknowledgments**

The authors gratefully acknowledge the partial financial support of this work by the German Research Foundation through the SAFARI project (grant no. 458030766) and the SFB 1608 (grant no. 501798263), the Research Council of Norway (grant no. 300172, SARLEM), and the research training group ConVeY, funded by the German Research Foundation under grant GRK 2428/2. Furthermore, we thank Jonathan Külz for his ideas on how to visualize the action aliasing phenomenon for the different learning paradigms.

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

# A  Appendix

## A.1  Action Projection Using Zonotopes

One option for defining safe action sets is to consider control invariant sets as safe state sets $\mathbb{X}^{\varphi}$, where $\boldsymbol{x}_t \in \mathbb{X}^{\varphi}$ ensures that there exists an admissible action $\boldsymbol{u}_t \in \mathbb{U}$ such that equation 10 can be satisfied for all times. The constraints in equation 12b can be defined using $\mathbb{C} \subseteq \mathbb{X}^{\varphi}$, where $\mathbb{C}$ is the set of all reachable states at the next time step under the system dynamics and bounded noise $\boldsymbol{w}_t \in \mathbb{W}$. The precise formulation of these constraints depends on the chosen set representations for approximating reachable and safe sets. In this work, we adopt zonotopes to enable efficient computation. Consequently, we have to replace equation 12b with the necessary constraints for verifying zonotope-in-zonotope containment. Consider a zonotope $\mathcal{Z} \subset \mathbb{R}^{n_z}$ in generator representation that is given by

$$\mathcal{Z} = \{\boldsymbol{z} \in \mathbb{R}^{n_z} : \boldsymbol{z} = \boldsymbol{c} + \boldsymbol{B}\boldsymbol{\beta}, |\boldsymbol{\beta}| \leq 1\} \tag{28}$$

with the center $\boldsymbol{c} \in \mathbb{R}^{n_z}$ and the generator matrix $\boldsymbol{B} \in \mathbb{R}^{n_z \times \eta(\mathcal{Z})}$, where $\eta(\mathcal{Z}) \in \mathbb{N}_0$ denotes the number of generators of $\mathcal{Z}$. The inequality in equation 28 has to be applied elementwise. A more compact notation is given by $\mathcal{Z} = \langle \boldsymbol{c}, \boldsymbol{B} \rangle_{\mathcal{Z}}$. Consider two zonotopes $\mathcal{Z}_1 = \langle \boldsymbol{c}_1, \boldsymbol{B}_1 \rangle_{\mathcal{Z}}$, $\mathcal{Z}_2 = \langle \boldsymbol{c}_2, \boldsymbol{B}_2 \rangle_{\mathcal{Z}}$. Then, $\mathcal{Z}_1$ is contained in $\mathcal{Z}_2$, i.e., $\mathcal{Z}_1 \subseteq \mathcal{Z}_2$, if there exist $\boldsymbol{\Gamma} \in \mathbb{R}^{\eta(\mathcal{Z}_2) \times \eta(\mathcal{Z}_1)}$, $\boldsymbol{\omega} \in \mathbb{R}^{\eta(\mathcal{Z}_2)}$ such that (Ghasemi et al., 2020, Lemma 1)

$$\boldsymbol{B}_1 = \boldsymbol{B}_2 \boldsymbol{\Gamma}, \tag{29a}$$

$$\boldsymbol{c}_2 - \boldsymbol{c}_1 = \boldsymbol{B}_2 \boldsymbol{\omega}, \tag{29b}$$

$$\| \begin{bmatrix} \boldsymbol{\Gamma} & \boldsymbol{\omega} \end{bmatrix} \|_{\infty} \leq 1. \tag{29c}$$

The resulting optimization problem involves both equality and inequality constraints, which is an important consideration when analyzing the sensitivity of the solution, as discussed in appendix A.2.

If the safe action set is given directly as a zonotope $\mathbb{U}_{\boldsymbol{x}}^{\varphi} = \langle \boldsymbol{c}_u, \boldsymbol{B}_u \rangle_{\mathcal{Z}}$, we only have to verify that $\tilde{\boldsymbol{u}} \in \mathbb{U}_{\boldsymbol{x}}^{\varphi}$. This can be achieved by replacing equation 12b with (Kulmburg & Althoff, 2021)

$$\boldsymbol{B}_u \boldsymbol{\nu} = \tilde{\boldsymbol{u}} - \boldsymbol{c}_u \tag{30}$$

$$\|\boldsymbol{\nu}\|_{\infty} \leq 1. \tag{31}$$

## A.2 Differentiating the Safeguard Using the Implicit Function Theorem

If the projection safeguard $\Phi$ is integrated into the policy as shown in figure 1b, we require the sensitivity of its output (the safe action) with respect to its input (the unsafe action) for the backward pass of the policy optimization. To obtain the sensitivity of $\boldsymbol{u}^{\varphi} = \Phi(\boldsymbol{x}, \boldsymbol{u})$ with respect to $\boldsymbol{u}$, let us first consider the Lagrange function associated with the projection problem 12,

$$\mathcal{L}(\boldsymbol{x}, \boldsymbol{u}, \tilde{\boldsymbol{u}}, \boldsymbol{\kappa}, \boldsymbol{v}) = \Omega(\boldsymbol{x}, \boldsymbol{u}, \tilde{\boldsymbol{u}}) + \boldsymbol{\kappa}^T D(\boldsymbol{x}, \boldsymbol{u}, \tilde{\boldsymbol{u}}) + \boldsymbol{v}^T H(\boldsymbol{x}, \boldsymbol{u}, \tilde{\boldsymbol{u}}),$$

where $\Omega$ is the safeguard objective, and the equality and inequality constraints resulting from the zonotope containment problem in equation 29 are represented with $D$ and $H$. The corresponding dual variables are $\boldsymbol{\kappa}$ and $\boldsymbol{v}$, respectively. Let us further consider the Karush-Kuhn-Tucker (KKT) conditions

$$\varepsilon(\boldsymbol{x}, \boldsymbol{u}, \tilde{\boldsymbol{u}}, \boldsymbol{\kappa}, \boldsymbol{v}) = \begin{bmatrix} \nabla_{\tilde{\boldsymbol{u}}} \mathcal{L}(\boldsymbol{x}, \boldsymbol{u}, \tilde{\boldsymbol{u}}, \boldsymbol{\kappa}, \boldsymbol{v}) \\ D(\boldsymbol{x}, \boldsymbol{u}, \tilde{\boldsymbol{u}}) \\ \boldsymbol{v}^T H(\boldsymbol{x}, \boldsymbol{u}, \tilde{\boldsymbol{u}}) \end{bmatrix}. \tag{32}$$

At the KKT point, we have

$$\varepsilon(\boldsymbol{x}, \boldsymbol{u}, \tilde{\boldsymbol{u}}, \boldsymbol{\kappa}, \boldsymbol{v})|_{\tilde{\boldsymbol{u}}=\boldsymbol{u}^{\varphi}, \boldsymbol{\kappa}=\boldsymbol{\kappa}^{\star}, \boldsymbol{v}=\boldsymbol{v}^{\star}} = \boldsymbol{0},$$

where $\{\boldsymbol{u}^{\varphi}, \boldsymbol{\kappa}^{\star}, \boldsymbol{v}^{\star}\}$ are the primal-dual solution of the safeguard. Then, with the implicit function theorem (Krantz & Parks, 2002), the sensitivity of the safeguard with respect to $\boldsymbol{u}$ is

$$\nabla_{\boldsymbol{u}} \Phi(\boldsymbol{x}, \boldsymbol{u}) = \nabla_{\boldsymbol{u}^{\varphi}} \varepsilon(\boldsymbol{x}, \boldsymbol{u}, \tilde{\boldsymbol{u}}, \boldsymbol{\kappa}, \boldsymbol{v})^{-1} \nabla_{\boldsymbol{u}} \varepsilon(\boldsymbol{x}, \boldsymbol{u}, \tilde{\boldsymbol{u}}, \boldsymbol{\kappa}, \boldsymbol{v}) \mid_{\tilde{\boldsymbol{u}}=\boldsymbol{u}^{\varphi}, \boldsymbol{\kappa}=\boldsymbol{\kappa}^{\star}, \boldsymbol{v}=\boldsymbol{v}^{\star}}. \tag{33}$$

## A.3 Proof of Theorem 1: Equivalence of Optimal Value Functions

To prove theorem 1, let us first recall the definition of an optimal policy. A policy is called optimal if its expected return is greater than or equal to that of all other policies in all states. Therefore, all optimal policies satisfy the same optimal value function (Sutton & Barto, 2018)

$$v^*(\boldsymbol{x}) = v_{\pi^\star}(\boldsymbol{x}) \geq v_{\pi}(\boldsymbol{x}) \quad \forall \boldsymbol{x} \in \tilde{\mathbb{X}}, \pi \in \Pi. \tag{34}$$

If we can show that SE-RL and SP-RL share the same optimal value function, this implies that any optimal policy in either framework will yield the same expected return. Note that policies in both frameworks, $\pi$ and $\pi^{\perp}$, are parameterized by $\boldsymbol{\theta}$, though this dependence is omitted for readability.

To demonstrate the equivalence of the optimal value functions, we begin by comparing the conditional probability densities of SE-RL – namely, the state transition density $p_x^{\mathrm{SE}}$ and the reward density $p_r^{\mathrm{SE}}$ – with those of SP-RL, denoted $p_x^{\mathrm{SP}}$ and $p_r^{\mathrm{SP}}$. At the core of both approaches, we have a sequence of mappings,

$$\boldsymbol{x} \xrightarrow{\pi} \boldsymbol{u} \xrightarrow{\Phi} \boldsymbol{u}^{\varphi} \xrightarrow{p^{\mathrm{SP}}} (\boldsymbol{x}', r),$$

where each arrow indicates either sampling from a conditional distribution or applying the deterministic transformation $\Phi$. However, the mappings are grouped differently in SE-RL and SP-RL, resulting in the sequences

$$\text{SP-RL}: \ \boldsymbol{x} \xrightarrow{\pi^{\perp}} \boldsymbol{u}^{\varphi} \xrightarrow{p^{\mathrm{SP}}} (\boldsymbol{x}', r),$$

$$\text{SE-RL}: \ \boldsymbol{x} \xrightarrow{\pi} \boldsymbol{u} \xrightarrow{p^{\mathrm{SE}}} (\boldsymbol{x}', r).$$

Thus, the Markov processes resulting from policies $\pi$ and $\pi^\perp$ with the same $\boldsymbol{\theta}$ would have the same conditional probability densities,

$$p_x^{\mathrm{SE}}(\boldsymbol{x}' \,|\, \boldsymbol{x}) = p_x^{\mathrm{SP}}(\boldsymbol{x}' \,|\, \boldsymbol{x})\,, \tag{35}$$

$$p_r^{\mathrm{SE}}(r \,|\, \boldsymbol{x}) = p_r^{\mathrm{SP}}(r \,|\, \boldsymbol{x})\,. \tag{36}$$

Now, we can prove theorem 1.

*Proof.* In SE-RL, for a certain policy $\pi$, the state value function (equation 13b) is given as

$$v_\pi^{\mathrm{SE}}(\boldsymbol{x}) = \mathbb{E}_{\mathbf{u}_t \sim \pi,\, \mathbf{x}_t \sim p_x^{\mathrm{SE}}} \left[ \mathrm{g}_t \middle| \mathbf{x}_0 = \boldsymbol{x} \right] \tag{37}$$

$$\overset{(1)}{=} \mathbb{E}_{\mathbf{u}_t \sim \pi,\, \mathbf{x}_t \sim p_x^{\mathrm{SE}}} \left[ \sum_{k=0}^\infty \gamma^k \mathrm{r}_{t+k+1} \middle| \mathbf{x}_0 = \boldsymbol{x} \right] \tag{38}$$

$$\overset{(a)}{=} \sum_{k=0}^\infty \gamma^k \int_{\tilde{\mathbb{X}}} \cdots \int_{\tilde{\mathbb{X}}} \int_{\mathbb{U}} \cdots \int_{\mathbb{U}} \int_{\mathbb{R}} r\, p_r^{\mathrm{SE}}(r \,|\, \boldsymbol{x}_k, \boldsymbol{u}_k) \pi(\boldsymbol{u}_k \,|\, \boldsymbol{x}_k)$$

$$\left[ \prod_{i=0}^{k-1} p_x^{\mathrm{SE}}(\boldsymbol{x}_{i+1} \,|\, \boldsymbol{x}_i, \boldsymbol{u}_i) \pi(\boldsymbol{u}_i \,|\, \boldsymbol{x}_i) \right] dr\, d\boldsymbol{u}_0 ... d\boldsymbol{u}_k\, d\boldsymbol{x}_1 ... d\boldsymbol{x}_k \tag{39}$$

$$\overset{(\mathrm{LTT})}{=} \sum_{k=0}^\infty \gamma^k \int_{\tilde{\mathbb{X}}} \cdots \int_{\tilde{\mathbb{X}}} \int_{\mathbb{R}} r\, p_r^{\mathrm{SE}}(r \,|\, \boldsymbol{x}_k) \left[ \prod_{i=0}^{k-1} p_x^{\mathrm{SE}}(\boldsymbol{x}_{i+1} \,|\, \boldsymbol{x}_i) \right] dr\, d\boldsymbol{x}_1 ... d\boldsymbol{x}_k\,, \tag{40}$$

where LTT refers to the law of total probability (Schervish, 1995, theorem B.70). Here, $(a)$ subsumes the linearity of expectation as well as the chain rule of probability with the Markov property. A similar derivation can be found in Bertsekas (2018, appendix A.2).

Similarly, for the corresponding safe policy $\pi^\perp$, we have

$$v_{\pi^\perp}^{\mathrm{SP}}(\boldsymbol{x}) = \mathbb{E}_{\mathbf{u}_t^\varphi \sim \pi^\perp,\, \mathbf{x}_t \sim p_x^{\mathrm{SP}}} \left[ \mathrm{g}_t \middle| \mathbf{x}_0 = \boldsymbol{x} \right] \tag{41}$$

$$\overset{(1)}{=} \mathbb{E}_{\mathbf{u}_t^\varphi \sim \pi^\perp,\, \mathbf{x}_t \sim p_x^{\mathrm{SP}}} \left[ \sum_{k=0}^\infty \gamma^k \mathrm{r}_{t+k+1} \middle| \mathbf{x}_0 = \boldsymbol{x} \right] \tag{42}$$

$$\overset{(a)}{=} \sum_{k=0}^\infty \gamma^k \int_{\tilde{\mathbb{X}}} \cdots \int_{\tilde{\mathbb{X}}} \int_{\mathbb{U}^\varphi} \cdots \int_{\mathbb{U}^\varphi} \int_{\mathbb{R}} r\, p_r^{\mathrm{SP}}(r \,|\, \boldsymbol{x}_k, \boldsymbol{u}_k^\varphi) \pi^\perp(\boldsymbol{u}_k^\varphi \,|\, \boldsymbol{x}_k)$$

$$\left[ \prod_{i=0}^{k-1} \pi^\perp(\boldsymbol{u}_i^\varphi \,|\, \boldsymbol{x}_i) p_x^{\mathrm{SP}}(\boldsymbol{x}_{i+1} \,|\, \boldsymbol{x}_i, \boldsymbol{u}_i^\varphi) \right] dr\, d\boldsymbol{u}_0^\varphi ... d\boldsymbol{u}_k^\varphi\, d\boldsymbol{x}_1 ... d\boldsymbol{x}_k \tag{43}$$

$$\overset{(\mathrm{LTT})}{=} \sum_{k=0}^\infty \gamma^k \int_{\tilde{\mathbb{X}}} \cdots \int_{\tilde{\mathbb{X}}} \int_{\mathbb{R}} r\, p_r^{\mathrm{SP}}(r \,|\, \boldsymbol{x}_k) \left[ \prod_{i=0}^{k-1} p_x^{\mathrm{SP}}(\boldsymbol{x}_{i+1} \,|\, \boldsymbol{x}_i) \right] dr\, d\boldsymbol{x}_1 ... d\boldsymbol{x}_k\,. \tag{44}$$

Thus, with equation 35 and equation 36, for a policy $\pi \in \Pi$, there exists a safe policy $\pi^\perp \in \Pi^\perp$ such that,

$$v_\pi^{\mathrm{SE}}(\boldsymbol{x}) = v_{\pi^\perp}^{\mathrm{SP}}(\boldsymbol{x}) \quad \forall\, \boldsymbol{x} \in \tilde{\mathbb{X}}\,. \tag{45}$$

Consider an optimal policy $\pi^\star$ for SE-RL, then the corresponding value function is the optimal value function in SE-RL, i.e. $v_{\pi^\star}^{\mathrm{SE}}(\boldsymbol{x}) = v^{\mathrm{SE}^\star}(\boldsymbol{x}), \forall \boldsymbol{x} \in \tilde{\mathbb{X}}$. Using equation 45, there exists a safe policy $\bar{\pi}^\perp \in \Pi^\perp$ such that $v_{\pi^\star}^{\mathrm{SE}}(\boldsymbol{x}) = v_{\bar{\pi}^\perp}^{\mathrm{SP}}(\boldsymbol{x}), \forall \boldsymbol{x} \in \tilde{\mathbb{X}}$. However, it is not guaranteed that this safe policy is optimal in the MDP $M^{\mathrm{SP}}$. Next, we employ a proof by contradiction to establish that the safe policy $\bar{\pi}^\perp$ and its associated state-value function $v_{\bar{\pi}^\perp}^{\mathrm{SP}}$ constitute the optimal policy and optimal state-value function for SP-RL.

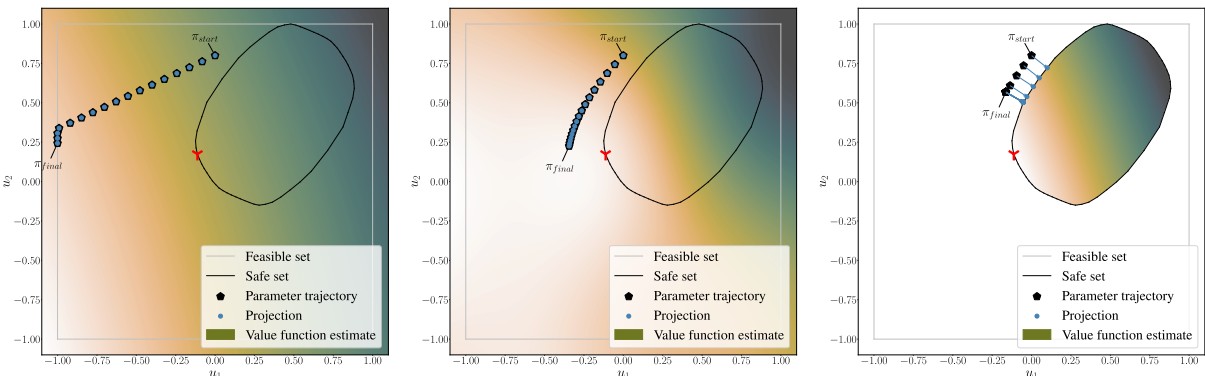

(a) Without safeguarding, the critic approximates the true objective function, showing that the optimal action would be unsafe. Note that we clip the policy to the feasible action set during policy improvement.

(b) SE-RL: Policy converges to an unsafe action that lies in the direction normal to the safe optimal action and the safe set boundary.

(c) SP-RL: Policy with differentiable projection safeguard does not improve in the direction normal to the projection and eventually gets stuck on a vertex of the safe action set.

Figure 6: Effect of action aliasing on SE-RL and SP-RL algorithms using deterministic policies. We illustrate the policy improvement step for a given state $\boldsymbol{x}$ in the navigation task. The deterministic policy $\pi(\boldsymbol{x})$ is updated for 50 steps using a loss function based on the learned state-action value function $q(\boldsymbol{x}, \boldsymbol{u})$ in the SE-RL case and $q(\boldsymbol{x}, \boldsymbol{u}^\varphi)$ in the SP-RL case.

Suppose that there exists a safe stochastic policy $\tilde{\pi}^\perp \in \Pi^\perp$ achieving strictly better performance,

$$v_{\tilde{\pi}^\perp}^{\mathrm{SP}}(\boldsymbol{x}) > v_{\bar{\pi}^\perp}^{\mathrm{SP}}(\boldsymbol{x}), \quad \text{for some } \boldsymbol{x} \in \tilde{\mathbb{X}}.$$

Since $\tilde{\pi}^\perp(\cdot \mid \boldsymbol{x})$ is supported on $\mathbb{U}^\varphi \subseteq \mathbb{U}$, we can construct a corresponding SE-RL policy $\tilde{\pi}(\cdot \mid \boldsymbol{x})$ whose pushforward through $\Phi$ equals $\tilde{\pi}^\perp(\cdot \mid \boldsymbol{x})$. Formally, for each $\boldsymbol{x}$:

$$\Phi_\# \tilde{\pi}(\cdot \mid \boldsymbol{x}) = \tilde{\pi}^\perp(\cdot \mid \boldsymbol{x}),$$

where $\Phi_\#$ denotes the pushforward of the probability measure under $\Phi$.

*Intuition:* for each safe action $\boldsymbol{u}^\varphi$ sampled from $\tilde{\pi}^\perp(\cdot \mid \boldsymbol{x})$, select a preimage $\boldsymbol{u} \in \Phi^{-1}(\boldsymbol{u}^\varphi)$ according to any probability distribution over that set. This is always possible because $\Phi$ is *surjective*, ensuring every $\boldsymbol{u}^\varphi \in \mathbb{U}^\varphi$ has at least one preimage in $\mathbb{U}$.

Using equation 45, the state-transition and reward distributions under $\tilde{\pi}$ in SE-RL match those under $\tilde{\pi}^\perp$ in SP-RL. Hence,

$$v_{\tilde{\pi}}^{\mathrm{SE}}(\boldsymbol{x}) = v_{\tilde{\pi}^\perp}^{\mathrm{SP}}(\boldsymbol{x}) > v_{\bar{\pi}^\perp}^{\mathrm{SP}}(\boldsymbol{x}) = v^{\mathrm{SE}^\star}(\boldsymbol{x}),$$

contradicting the optimality of $\pi^\star$.

Therefore, no such $\tilde{\pi}^\perp$ exists, and $\bar{\pi}^\perp$ is also optimal in SP-RL. Consequently, the optimal value functions coincide,

$$v^{\mathrm{SE}^\star}(\boldsymbol{x}) = v^{\mathrm{SP}^\star}(\boldsymbol{x}), \quad \forall \boldsymbol{x} \in \tilde{\mathbb{X}}.$$

$\square$

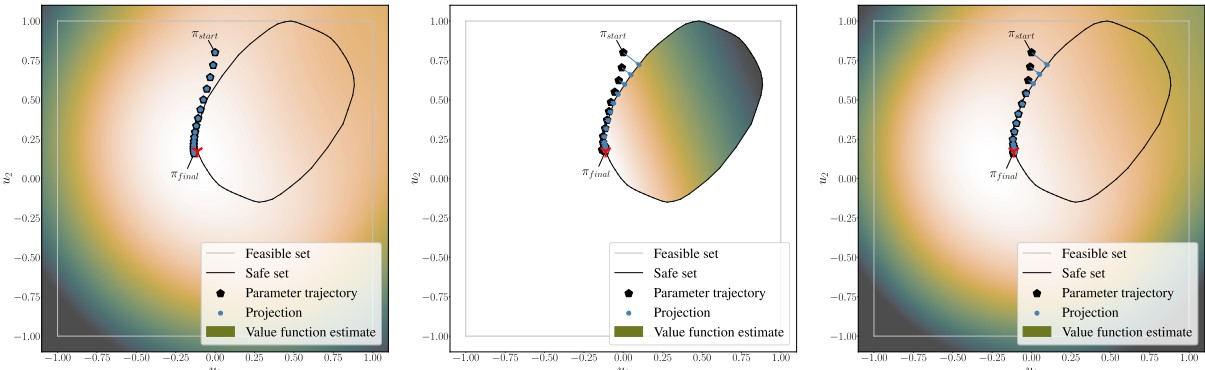

(a) SE-RL: Adding a penalty to the reward addresses the flat-lining critic problem and improves convergence toward the optimal safe action.

(b) SP-RL: An additional policy loss term that penalizes the distance between unsafe and safe action improves convergence to the optimal safe action. However, if the optimal safe action would lie in the interior of the safe action set, it might not be reached.

(c) SP-RL: Learning an additional penalty critic is very similar to adding a penalty to the reward in SE-RL. Note that as the penalty critic is conditioned on unsafe actions, we display the objective over the entire action space.

Figure 7: Effect of improvement strategies when using a differentiable safeguard during policy updates for a given state $\boldsymbol{x}$ in the navigation task. The deterministic policy $\pi_{\boldsymbol{\theta}}(\boldsymbol{x})$ is updated for 50 steps using a loss function based on the learned state-action value function $q(\boldsymbol{x}, \boldsymbol{u})$ in the SE-RL case and $q(\boldsymbol{x}, \boldsymbol{u}^{\varphi})$ in the SP-RL case.

## A.4 Proof of Lemma 1: Equivalence of SE-RL and SP-RL for Stochastic Policies and GAE

*Proof.* With $\Psi_{\pi_{\perp}}^{\mathrm{SP}}(\boldsymbol{x}, \boldsymbol{u}^{\varphi}) = \hat{a}^{\mathrm{GAE}}(\boldsymbol{x}, \boldsymbol{u}^{\varphi})$ and $\Psi_{\pi_{\perp}}^{\mathrm{SE}}(\boldsymbol{x}, \boldsymbol{u}) = \hat{a}^{\mathrm{GAE}}(\boldsymbol{x}, \boldsymbol{u})$, we can rewrite the policy gradient estimates for SP-RL for stochastic policies as

$$
\begin{aligned}
\nabla_{\boldsymbol{\theta}} J(\pi_{\boldsymbol{\theta}}^{\perp}) &= \mathbb{E}_{\pi^{\perp}} \left[ \Psi_{\pi_{\perp}}^{\mathrm{SP}}(\boldsymbol{x}, \boldsymbol{u}^{\varphi}) \nabla_{\boldsymbol{\theta}} \log \pi_{\boldsymbol{\theta}}(\boldsymbol{u} \,|\, \boldsymbol{x}) \right] \\
&= \mathbb{E}_{\pi^{\perp}} \left[ \hat{a}_{\pi_{\perp}}^{\mathrm{GAE}}(\boldsymbol{x}, \boldsymbol{u}^{\varphi}) \nabla_{\boldsymbol{\theta}} \log \pi_{\boldsymbol{\theta}}(\boldsymbol{u} \,|\, \boldsymbol{x}) \right] \\
&\overset{(7)}{=} \mathbb{E}_{\pi^{\perp}} \left[ \sum_{l=0}^{\infty} (\gamma\lambda)^{l} \delta_{t+l}^{v} \nabla_{\boldsymbol{\theta}} \log \pi_{\boldsymbol{\theta}}(\boldsymbol{u} \,|\, \boldsymbol{x}) \right] \\
&\overset{(8)}{=} \mathbb{E}_{\pi^{\perp}} \left[ \sum_{l=0}^{\infty} (\gamma\lambda)^{l} (r_{t+l} + \gamma v_{\pi_{\perp}}^{\mathrm{SP}}(\boldsymbol{x}_{t+l+1}) - v_{\pi_{\perp}}^{\mathrm{SP}}(\boldsymbol{x}_{t+l})) \nabla_{\boldsymbol{\theta}} \log \pi_{\boldsymbol{\theta}}(\boldsymbol{u} \,|\, \boldsymbol{x}) \right]
\end{aligned}
$$

and, similarly,

$$
\nabla_{\boldsymbol{\theta}} J(\pi_{\boldsymbol{\theta}}) = \mathbb{E}_{\pi} \left[ \sum_{l=0}^{\infty} (\gamma\lambda)^{l} (r_{t+l} + \gamma v_{\pi}^{\mathrm{SE}}(\boldsymbol{x}_{t+l+1}) - v_{\pi}^{\mathrm{SE}}(\boldsymbol{x}_{t+l})) \nabla_{\boldsymbol{\theta}} \log \pi_{\boldsymbol{\theta}}(\boldsymbol{u} \,|\, \boldsymbol{x}) \right].
$$

Using the same initial parameters $\boldsymbol{\phi} = \boldsymbol{\phi}^{\mathrm{in}}$ and $\boldsymbol{\theta} = \boldsymbol{\theta}^{\mathrm{in}}$, both policy gradient estimates are the same for the first policy update. Similarly, as the value functions $v_{\pi}^{\mathrm{SE}}$ and $v_{\pi_{\perp}}^{\mathrm{SP}}$ only depend on the state, the gradient $\nabla_{\boldsymbol{\phi}} L_{\boldsymbol{\phi}}$ for the loss defined in equation 9 is also equivalent. Consequently, if all other algorithm hyperparameters are the same for both SE-RL and SP-RL, all subsequent parameter updates will also be the same. □

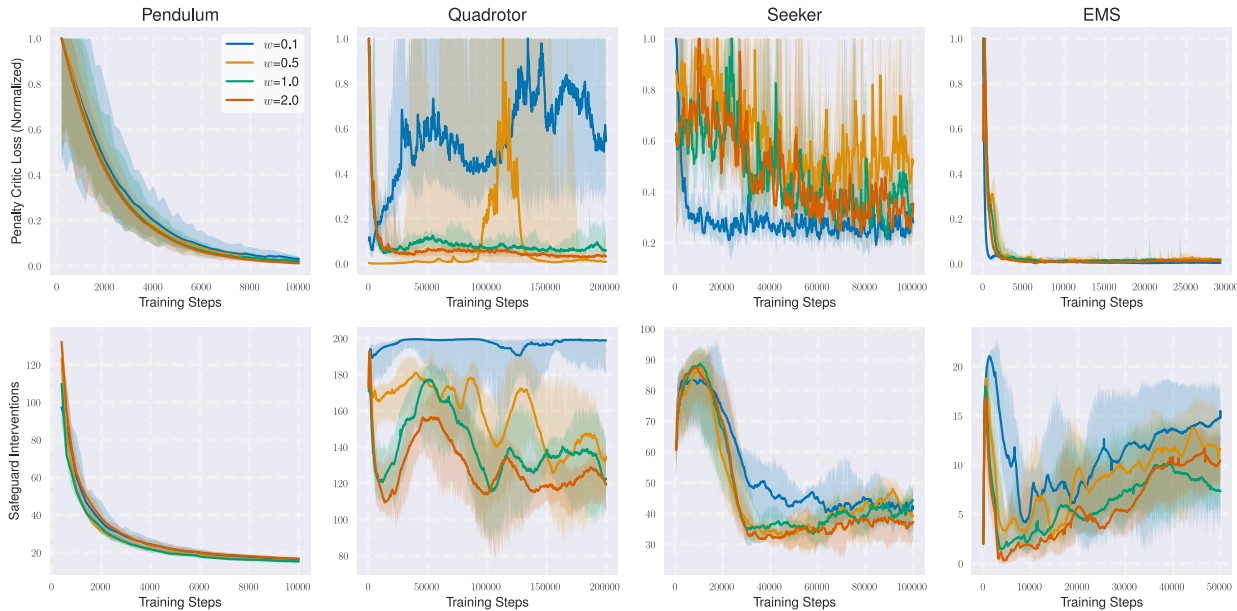

Figure 8: Comparison of the penalty critic loss and the number of safeguard interventions during training for different scaling factors $w$.

## A.5 Proof of Lemma 3: Flat-lining Critic in SE-RL

*Proof.* Any action $\boldsymbol{u}^e \in \mathbb{U}^e(\boldsymbol{u}^b)$ is projected to $\boldsymbol{u}^b$ and thus transitions to the same next $\boldsymbol{x}'$ and yields the same reward $r^b \sim p_r(r \,|\, \boldsymbol{x}, \boldsymbol{u}^b)$. Since the state-action value function satisfies (Sutton & Barto, 2018)

$$q_\pi(\boldsymbol{x}, \boldsymbol{u}) = \mathbb{E}\left[r + \gamma v_\pi(\boldsymbol{x}') \,|\, \mathbf{x}_t = \boldsymbol{x}, \mathbf{u}_t = \boldsymbol{u}\right],$$

we obtain

$$q_\pi(x, \boldsymbol{u}^e) = q_\pi(\boldsymbol{x}, \boldsymbol{u}^b) \quad \forall \boldsymbol{u}^e \in \mathbb{U}^e,$$

and, consequently,

$$a_\pi(\boldsymbol{x}, \boldsymbol{u}^e) = a_\pi(\boldsymbol{x}, \boldsymbol{u}^b) \quad \forall \boldsymbol{u}^e \in \mathbb{U}^e.$$

Furthermore, for the GAE according to equation 7, we receive

$$\hat{a}_\pi^{\mathrm{GAE}}(\boldsymbol{x}, \boldsymbol{u}^e) = \hat{a}_\pi^{\mathrm{GAE}}(\boldsymbol{x}, \boldsymbol{u}^b) \quad \forall \boldsymbol{u}^e \in \mathbb{U}^e.$$

$\square$

## A.6 Action Aliasing Visualization for Seeker Task

We provide a second minimal example to visualize the impact of action aliasing in deterministic policies. It is based on the seeker navigation task described in appendix A.8. All other settings are the same as listed in section 6.3. The main difference to the previously described example is that task performance and safety are not aligned in the navigation task, as one obstacle will always be situated between the initial and the goal position. Therefore, actions that are optimal with respect to the task performance are usually unsafe, and the optimal safe action lies on the boundary of the safe action set. This is visualized in figure 6. Due to these different task characteristics, the impact of the different improvement strategies becomes more similar as shown in figure 7.

## A.7 Details of Penalty Critic Implementation

The penalty critic is a separate neural network with the same architecture as the task critic. In contrast to the latter, the penalty critic is conditioned on the unsafe action $\boldsymbol{u}$, not the safe action. Its parameters $\varsigma$ are

trained using the mean-squared error loss in equation 9, where the target $y$ is computed as

$$y_t = \underbrace{w\|\boldsymbol{u}_t - \boldsymbol{u}_t^\varphi\|_2^2}_{h_t} + \gamma q_{\pi,\bar{\boldsymbol{\varsigma}}}^{\text{pen}}(\boldsymbol{x}_{t+1}, \boldsymbol{u}_{t+1}). \tag{46}$$

Consequently, the scaling factor $w$ has a strong impact on the learning stability of the penalty critic. Figure 8 shows that while for the pendulum and the EMS all penalty factors lead to a stable loss progression and reduced safeguard interventions, only higher values for $w$ can achieve this for the quadrotor environment. For the seeker navigation task, where performance and safety are not aligned (see section 8.1), a low penalty factor enables the best convergence behavior.

### A.8 Benchmark Problems

### A.8.1 Pendulum Stabilization Task

Our pendulum environment is closely related to the *OpenAI Gym Pendulum-V0*[3] environment with the difference that we limit the one-dimensional control input to $|u| \leq 8\,\text{rads}^{-1}$. The environment has the state $\boldsymbol{x} = \left[\vartheta, \dot{\vartheta}\right]^T$ and the dynamics

$$\dot{\boldsymbol{x}} = \begin{pmatrix} \dot{\vartheta} \\ \frac{g}{\ell}\sin(\vartheta) + \frac{1}{m\ell^2}u \end{pmatrix}, \tag{47}$$

where $g$ is gravity and $m, \ell$ are the mass and the length of the pendulum, respectively. We discretize the dynamics using the explicit Euler method. The desired equilibrium state is $\boldsymbol{x}^* = \left[0, 0\right]^T$. The reward is $r = -(\vartheta^2 + 0.1\dot{\vartheta}^2 + 0.001u^2)$.

### A.8.2 Quadrotor Stabilization Task

We use the model of a quadrotor operating in the $x$-$z$-plane that is proposed in (Stolz et al., 2024). The system has two independent thrusts $\boldsymbol{u} = \left[u^1, u^2\right]^T$ bounded by lower and upper limits $\underline{\boldsymbol{u}}, \overline{\boldsymbol{u}}$, respectively. The state of the system is $\boldsymbol{x} = \left[e^x, e^z, \dot{e}^x, \dot{e}^z, \vartheta, \dot{\vartheta}\right]^T$, where $e_{x,z}$ are the positions along the $x$- and $z$-axis, respectively. The system dynamics are

$$\dot{\boldsymbol{x}} = \begin{pmatrix} \dot{e}^x \\ \dot{e}^z \\ (u^1 + u^2)K\sin(\vartheta) \\ -g + (u^1 + u^2)K\cos(\vartheta) \\ \dot{\vartheta} \\ -d_0\vartheta - d_1\dot{\vartheta} + n_0(-u^1 + u^2) \end{pmatrix} + \begin{pmatrix} 0 \\ 0 \\ w^1 \\ w^2 \\ 0 \\ 0 \end{pmatrix}, \tag{48}$$

where $d_0, d_1, n_0, K$ are constants and $\boldsymbol{w} = \left[w^1, w^2\right]^T$ are disturbances. The linearized dynamics are obtained using a first-order Taylor expansion around the equilibrium point $\boldsymbol{x}^* = \left[0, 1, 0, 0, 0, 0\right]^T$ and are discretized in time using the explicit Euler method. The disturbances are sampled uniformly from a compact disturbance set $\mathbb{W} \subset \mathbb{R}^2$. The reward is computed using $r = -1 + \exp\left(-\|\boldsymbol{s} - \boldsymbol{s}^*\|_2 - \frac{0.01}{2}\|\left[\frac{u^1 - \underline{u}^1}{\overline{u}^1 - \underline{u}^1}, \frac{u^2 - \underline{u}^2}{\overline{u}^2 - \underline{u}^2}\right]\|_1\right)$, where $\boldsymbol{s} = \left[e^x, e^z\right]$.

### A.8.3 Seeker Navigation Task

In this two-dimensional navigation task, a simple massless robot has to find the shortest path to the goal while avoiding a fixed number of obstacles. The environment is configured such that at least one obstacle always lies between the initial position of the seeker and the goal position $\boldsymbol{e}^g$. The initial position, goal position, and the positions and radii of the obstacles are pseudo-randomly sampled at the beginning of each episode. The state $\boldsymbol{x} = [e^x, e^y, \dot{e}^x, \dot{e}^y]^T$ consists of the positions and velocities in the $x$-$y$-plane. The

---

[3]https://gymnasium.farama.org/environments/classic_control/pendulum/

actions $\boldsymbol{u} = [\ddot{e}^x, \ddot{e}^y]$ are accelerations in the respective directions and bounded by $|\ddot{e}^x|, |\ddot{e}^y| \leq a^{\max}$ with $a^{\max} = 1\,m/s^2$. We use the simplified system dynamics

$$\dot{\boldsymbol{x}} = \begin{pmatrix} \dot{e}^x \\ \dot{e}^y \\ \ddot{e}^x \\ \ddot{e}^y \end{pmatrix} + \begin{pmatrix} 0 \\ 0 \\ w^1 \\ w^2 \end{pmatrix}, \tag{49}$$

and discretize them using the explicit Euler method. The disturbances $\boldsymbol{w} = \begin{bmatrix} w^1, w^2 \end{bmatrix}^T$ are uniformly sampled from a compact disturbance set $\mathbb{W} \subset \mathbb{R}^2$. The position of the seeker at time $t$ is $\boldsymbol{e}_t$, and the step reward is $r = -1 + \exp(\|\boldsymbol{e}_t - \boldsymbol{e}_g\|)$. In all experiments, we set the number of obstacles to three.

In each time step, a safe action set $\mathbb{U}_{\boldsymbol{x}}^{\varphi}(x)$ is computed to avoid collisions with obstacles and meet the boundaries of the map. This simplifies the projection problem in equation 12 to

$$\Phi(\boldsymbol{x}, \boldsymbol{u}):$$
$$\boldsymbol{u}^{\varphi} = \arg\min_{\tilde{\boldsymbol{u}}} \frac{1}{2}\|\tilde{\boldsymbol{u}} - \boldsymbol{u}\|_2^2 \tag{50}$$
$$\text{s.t.} \quad \tilde{\boldsymbol{u}} \in \mathbb{U}_{\boldsymbol{x}}^{\varphi}. \tag{51}$$
$$\tag{52}$$

### A.8.4 Energy Management System Optimization Task

The goal in this task (introduced in Walter et al. (2025)) is to control a battery and a heat pump in a building that has a non-flexible electric base load and a non-controllable PV generator. The state $\boldsymbol{x} = [soc, \Theta^{\text{in}}, \Theta^{\text{ret}}]^T$ comprises the state of charge of the battery, the indoor temperature, and the return temperature of the floor heating system. The policy of the agent is conditioned on the observation $\boldsymbol{o} = [\boldsymbol{x}, \Theta_{t:t+H}^{\text{out}}, p_{t:t+H}^{\text{PV}}, p_{t:t+H}^{\text{L}}, \varrho_{t:t+H}]$, which is composed of the current value and $H$ forecasts of the outdoor temperature, the PV power, the inflexible load, and the electricity price. We use $H = 5$ such that the observation space has a dimensionality of 23. The actions $\boldsymbol{u} = [p^{\text{B}}, p^{\text{HP}}]$ are the continuous power set points for the battery and the heat pump, respectively. The system follows the simplified dynamics

$$\dot{\boldsymbol{x}} = \begin{pmatrix} p^{\text{B}} \\ -c_0\Theta^{\text{in}} + c_1\Theta^{\text{ret}} \\ c_2\Theta^{\text{in}} - c_2\Theta^{\text{ret}} + c_3 p^{\text{HP}} \end{pmatrix} + \begin{pmatrix} 0 \\ c_4\Theta^{\text{out}} \\ 0 \end{pmatrix}, \tag{53}$$

where the outdoor temperature acts as a bounded noise. The computation of the coefficients $c_{0-4}$ is detailed in Bianchi (2006, equation 2.17). Performance is measured through the deviation of the indoor temperature from a desired set point $\Theta^{\text{set}}$ as well as through the electricity costs such that $r = -(p^{\text{B}} + p^{\text{HP}} + p^{\text{L}} - p^{\text{PV}})\varrho - (\Theta^{\text{in}} - \Theta^{\text{set}})^2$. The state constraints $\mathbb{X} = [0, 10] \times [18, 24] \times [10, 100]$ are enforced using reachability analysis. During training, one episode corresponds to one day, and the initial state is randomized.

### A.9 Wall Clock Time Results

Table 2: Comparison of wall clock time for TD3 (deterministic policy case) over 10,000 training steps, normalized to the vanilla SE-RL approach.

| | States | SE-RL | SE-RL Penalty | SP-RL | SP-RL PSL | SP-RL PenC |
|---|---|---|---|---|---|---|
| **Pendulum** | 2 | 1.0 | 0.99 | 6.04 | 5.59 | 5.49 |
| **Seeker** | 2 | 1.0 | 1.0 | 3.56 | 3.55 | 3.71 |
| **Quadrotor** | 6 | 1.0 | 1.0 | 12.01 | 10.51 | 9.02 |
| **EMS** | 3 | 1.0 | 0.98 | 12.25 | 12.38 | 12.51 |

### A.10   Comprehensive Results

Table 3: TD3, pendulum: mean and standard deviation of returns and safeguard interventions for different scaling factors $w$.

| | SE-RL | SE-RL Penalty | SP-RL | SP-RL PSL | SP-RL PenC |
|---|---|---|---|---|---|
| | | | $w = 0.1$ | | |
| **Interventions** | $0.01 \pm 0.20$ | $0.03 \pm 0.24$ | $4.13 \pm 5.10$ | $1.33 \pm 1.35$ | $5.49 \pm 18.23$ |
| **Returns** | $-11.52 \pm 7.24$ | $-13.23 \pm 8.19$ | $-10.52 \pm 7.41$ | $-8.06 \pm 6.16$ | $-8.71 \pm 6.22$ |
| | | | $w = 0.5$ | | |
| **Interventions** | $0.01 \pm 0.20$ | $0.0 \pm 0.0$ | $4.13 \pm 5.10$ | $2.51 \pm 4.17$ | $8.90 \pm 18.29$ |
| **Returns** | $-11.52 \pm 7.24$ | $-29.34 \pm 19.43$ | $-10.52 \pm 7.41$ | $-8.77 \pm 6.21$ | $-7.60 \pm 5.99$ |
| | | | $w = 1.0$ | | |
| **Interventions** | $0.01 \pm 0.20$ | $0.07 \pm 0.35$ | $4.13 \pm 5.10$ | $1.36 \pm 1.46$ | $13.06 \pm 27.32$ |
| **Returns** | $-11.52 \pm 7.24$ | $-23.26 \pm 21.19$ | $-10.52 \pm 7.41$ | $-8.33 \pm 6.84$ | $-9.57 \pm 7.08$ |
| | | | $w = 2.0$ | | |
| **Interventions** | $0.01 \pm 0.20$ | $0.0 \pm 0.0$ | $4.13 \pm 5.10$ | $18.57 \pm 34.74$ | $16.53 \pm 25.80$ |
| **Returns** | $-11.52 \pm 7.24$ | $-45.48 \pm 23.71$ | $-10.52 \pm 7.41$ | $-8.50 \pm 6.30$ | $-7.47 \pm 5.70$ |

Table 4: TD3, quadrotor: mean and standard deviation of returns and safeguard interventions for different scaling factors $w$.

| | SE-RL | SE-RL Penalty | SP-RL | SP-RL PSL | SP-RL PenC |
|---|---|---|---|---|---|
| | | | $w = 0.1$ | | |
| **Interventions** | $194.23 \pm 14.01$ | $170.13 \pm 48.93$ | $199.71 \pm 0.64$ | $199.04 \pm 1.49$ | $192.60 \pm 23.79$ |
| **Returns** | $-49.43 \pm 6.76$ | $-49.75 \pm 12.35$ | $-66.34 \pm 16.76$ | $-66.87 \pm 17.41$ | $-62.52 \pm 13.00$ |
| | | | $w = 0.5$ | | |
| **Interventions** | $194.23 \pm 14.01$ | $64.16 \pm 46.66$ | $199.71 \pm 0.64$ | $170.44 \pm 51.05$ | $145.31 \pm 49.21$ |
| **Returns** | $-49.43 \pm 6.76$ | $-43.76 \pm 13.68$ | $-66.34 \pm 16.76$ | $-59.11 \pm 18.65$ | $-53.46 \pm 17.47$ |
| | | | $w = 1.0$ | | |
| **Interventions** | $194.23 \pm 14.01$ | $65.63 \pm 60.98$ | $199.71 \pm 0.64$ | $181.79 \pm 37.02$ | $141.80 \pm 58.84$ |
| **Returns** | $-49.43 \pm 6.76$ | $-43.33 \pm 16.68$ | $-66.34 \pm 16.76$ | $-49.97 \pm 7.68$ | $-54.22 \pm 16.73$ |
| | | | $w = 2.0$ | | |
| **Interventions** | $194.23 \pm 14.01$ | $45.94 \pm 40.76$ | $199.71 \pm 0.64$ | $192.51 \pm 21.70$ | $134.93 \pm 52.98$ |
| **Returns** | $-49.43 \pm 6.76$ | $-42.81 \pm 11.78$ | $-66.34 \pm 16.76$ | $-49.34 \pm 5.74$ | $-44.34 \pm 10.26$ |

Table 5: TD3, seeker: mean and standard deviation of returns and safeguard interventions for different scaling factors $w$.

|  | SE-RL | SE-RL Penalty | SP-RL | SP-RL PSL | SP-RL PenC |
|---|---|---|---|---|---|
|  |  |  | $w = 0.1$ |  |  |
| Interventions | $83.90 \pm 34.00$ | $85.60 \pm 32.19$ | $81.7 \pm 35.16$ | $42.24 \pm 43.90$ | $49.90 \pm 44.94$ |
| Returns | $-34.20 \pm 20.05$ | $-34.16 \pm 20.14$ | $-51.33 \pm 22.97$ | $-26.31 \pm 20.13$ | $-31.66 \pm 20.43$ |
|  |  |  | $w = 0.5$ |  |  |
| Interventions | $83.90 \pm 34.00$ | $80.00 \pm 37.10$ | $81.7 \pm 35.16$ | $44.01 \pm 39.53$ | $42.63 \pm 44.65$ |
| Returns | $-34.20 \pm 20.05$ | $-32.73 \pm 20.70$ | $-51.33 \pm 22.97$ | $-26.10 \pm 18.19$ | $-31.02 \pm 19.22$ |
|  |  |  | $w = 1.0$ |  |  |
| Interventions | $83.90 \pm 34.00$ | $93.96 \pm 22.04$ | $81.7 \pm 35.16$ | $43.19 \pm 38.87$ | $46.03 \pm 45.75$ |
| Returns | $-34.20 \pm 20.05$ | $-39.12 \pm 16.59$ | $-51.33 \pm 22.97$ | $-28.17 \pm 17.49$ | $-36.06 \pm 19.26$ |
|  |  |  | $w = 2.0$ |  |  |
| Interventions | $83.90 \pm 34.00$ | $97.47 \pm 14.86$ | $81.7 \pm 35.16$ | $40.40 \pm 39.20$ | $39.31 \pm 2.97$ |
| Returns | $-34.20 \pm 20.05$ | $-48.09 \pm 15.86$ | $-51.33 \pm 22.97$ | $-28.90 \pm 19.73$ | $-42.04 \pm 17.69$ |

Table 6: TD3, EMS: mean and standard deviation of returns and safeguard interventions for different scaling factors $w$.

|  | SE-RL | SE-RL Penalty | SP-RL | SP-RL PSL | SP-RL PenC |
|---|---|---|---|---|---|
|  |  |  | $w = 0.1$ |  |  |
| Interventions | $17.19 \pm 7.42$ | $3.1 \pm 4.36$ | $22.99 \pm 0.12$ | $16.23 \pm 4.46$ | $14.11 \pm 7.75$ |
| Returns | $-16.11 \pm 8.81$ | $-18.71 \pm 8.77$ | $-26.45 \pm 19.25$ | $-21.60 \pm 15.51$ | $-14.59 \pm 6.46$ |
|  |  |  | $w = 0.5$ |  |  |
| Interventions | $17.19 \pm 7.42$ | $0.43 \pm 0.79$ | $22.99 \pm 0.12$ | $13.9 \pm 8.78$ | $15.13 \pm 4.55$ |
| Returns | $-16.11 \pm 8.81$ | $-20.02 \pm 8.65$ | $-26.45 \pm 19.25$ | $-15.63 \pm 7.11$ | $-12.65 \pm 4.59$ |
|  |  |  | $w = 1.0$ |  |  |
| Interventions | $17.19 \pm 7.42$ | $0.0 \pm 0.0$ | $22.99 \pm 0.12$ | $15.81 \pm 4.94$ | $11.07 \pm 7.03$ |
| Returns | $-16.11 \pm 8.81$ | $-23.06 \pm 12.00$ | $-26.45 \pm 19.25$ | $-13.57 \pm 5.87$ | $-14.41 \pm 6.23$ |
|  |  |  | $w = 2.0$ |  |  |
| Interventions | $17.19 \pm 7.42$ | $0.0 \pm 0.0$ | $22.99 \pm 0.12$ | $8.56 \pm 7.47$ | $11.71 \pm 7.16$ |
| Returns | $-16.11 \pm 8.81$ | $-20.9 \pm 8.03$ | $-26.45 \pm 19.25$ | $-18.40 \pm 13.90$ | $-12.59 \pm 5.13$ |

Table 7: A2C, pendulum: mean and standard deviation of returns and safeguard interventions for different scaling factors $w$.

| | SE-RL | SE-RL Penalty | SP-RL | SP-RL PSL | SP-RL PenC |
|---|---|---|---|---|---|
| | | | $w = 0.1$ | | |
| **Interventions** | $170.03 \pm 53.11$ | $1.04 \pm 2.82$ | $170.03 \pm 53.11$ | $6.06 \pm 8.88$ | $1.04 \pm 2.82$ |
| **Returns** | $-177.09 \pm 31.28$ | $-32.90 \pm 57.86$ | $-177.09 \pm 31.28$ | $-7.60 \pm 6.33$ | $-32.90 \pm 57.86$ |
| | | | $w = 0.5$ | | |
| **Interventions** | $170.03 \pm 53.11$ | $0.48 \pm 2.15$ | $170.03 \pm 53.11$ | $7.84 \pm 11.95$ | $0.48 \pm 2.15$ |
| **Returns** | $-177.09 \pm 31.28$ | $-24.58 \pm 40.57$ | $-177.09 \pm 31.28$ | $-7.52 \pm 6.54$ | $-24.58 \pm 40.57$ |
| | | | $w = 1.0$ | | |
| **Interventions** | $170.03 \pm 53.11$ | $50.79 \pm 80.19$ | $170.03 \pm 53.11$ | $27.50 \pm 36.68$ | $50.79 \pm 80.19$ |
| **Returns** | $-177.09 \pm 31.28$ | $-114.15 \pm 77.99$ | $-177.09 \pm 31.28$ | $-8.02 \pm 7.40$ | $-114.15 \pm 77.99$ |
| | | | $w = 2.0$ | | |
| **Interventions** | $170.03 \pm 53.11$ | $38.66 \pm 68.81$ | $170.03 \pm 53.11$ | $11.66 \pm 20.81$ | $38.66 \pm 68.81$ |
| **Returns** | $-177.09 \pm 31.28$ | $-156.37 \pm 64.00$ | $-177.09 \pm 31.28$ | $-7.06 \pm 6.42$ | $-156.37 \pm 64.00$ |

Table 8: A2C, quadrotor: mean and standard deviation of returns and safeguard interventions for different scaling factors $w$.

| | SE-RL | SE-RL Penalty | SP-RL | SP-RL PSL | SP-RL PenC |
|---|---|---|---|---|---|
| | | | $w = 0.1$ | | |
| **Interventions** | $199.90 \pm 0.46$ | $161.56 \pm 50.16$ | $199.90 \pm 0.46$ | $198.27 \pm 4.67$ | $161.56 \pm 50.16$ |
| **Returns** | $-66.09 \pm 20.09$ | $-50.46 \pm 8.22$ | $-66.09 \pm 20.09$ | $-67.58 \pm 25.27$ | $-50.46 \pm 8.22$ |
| | | | $w = 0.5$ | | |
| **Interventions** | $199.90 \pm 0.46$ | $98.27 \pm 64.17$ | $199.90 \pm 0.46$ | $198.46 \pm 2.19$ | $98.27 \pm 64.17$ |
| **Returns** | $-66.09 \pm 20.09$ | $-52.75 \pm 12.63$ | $-66.09 \pm 20.09$ | $-58.60 \pm 9.41$ | $-52.75 \pm 12.63$ |
| | | | $w = 1.0$ | | |
| **Interventions** | $199.90 \pm 0.46$ | $63.96 \pm 63.01$ | $199.90 \pm 0.46$ | $196.94 \pm 4.97$ | $63.96 \pm 63.01$ |
| **Returns** | $-66.09 \pm 20.09$ | $-50.29 \pm 13.41$ | $-66.09 \pm 20.09$ | $-62.94 \pm 10.30$ | $-50.29 \pm 13.41$ |
| | | | $w = 2.0$ | | |
| **Interventions** | $199.90 \pm 0.46$ | $80.19 \pm 82.43$ | $199.90 \pm 0.46$ | $195.77 \pm 5.66$ | $80.19 \pm 82.43$ |
| **Returns** | $-66.09 \pm 20.09$ | $-51.52 \pm 14.81$ | $-66.09 \pm 20.09$ | $-63.81 \pm 12.06$ | $-51.52 \pm 14.81$ |

Table 9: A2C, seeker: mean and standard deviation of returns and safeguard interventions for different scaling factors $w$.

| | SE-RL | SE-RL Penalty | SP-RL | SP-RL PSL | SP-RL PenC |
|---|---|---|---|---|---|
| **Interventions** | $56.93 \pm 44.75$ | $66.42 \pm 44.02$ | $56.93 \pm 44.75$ | $43.71 \pm 42.01$ | $66.42 \pm 44.02$ |
| **Returns** | $-19.46 \pm 19.92$ | $-22.04 \pm 20.23$ | $-19.46 \pm 19.92$ | $-24.82 \pm 17.54$ | $-22.04 \pm 20.23$ |
| | | | $w = 0.5$ | | |
| **Interventions** | $56.93 \pm 44.75$ | $97.43 \pm 15.16$ | $56.93 \pm 44.75$ | $61.33 \pm 40.95$ | $97.43 \pm 15.16$ |
| **Returns** | $-19.46 \pm 19.92$ | $-41.33 \pm 15.46$ | $-19.46 \pm 19.92$ | $-35.58 \pm 13.76$ | $-41.33 \pm 15.46$ |
| | | | $w = 1.0$ | | |
| **Interventions** | $56.93 \pm 44.75$ | $100.00 \pm 0.00$ | $56.93 \pm 44.75$ | $72.61 \pm 34.33$ | $100.00 \pm 0.00$ |
| **Returns** | $-19.46 \pm 19.92$ | $-50.22 \pm 15.04$ | $-19.46 \pm 19.92$ | $-38.61 \pm 13.28$ | $-50.22 \pm 15.04$ |
| | | | $w = 2.0$ | | |
| **Interventions** | $56.93 \pm 44.75$ | $100.00 \pm 0.00$ | $56.93 \pm 44.75$ | $64.86 \pm 36.08$ | $100.00 \pm 0.00$ |
| **Returns** | $-19.46 \pm 19.92$ | $-56.06 \pm 14.32$ | $-19.46 \pm 19.92$ | $-40.67 \pm 13.33$ | $-56.06 \pm 14.32$ |

Table 10: A2C, EMS: mean and standard deviation of returns and safeguard interventions for different scaling factors $w$.

| | SE-RL | SE-RL Penalty | SP-RL | SP-RL PSL | SP-RL PenC |
|---|---|---|---|---|---|
| | | | $w = 0.1$ | | |
| **Interventions** | $17.49 \pm 1.67$ | $12.6 \pm 4.02$ | $17.49 \pm 1.67$ | $16.23 \pm 4.46$ | $10.87 \pm 5.51$ |
| **Returns** | $-14.14 \pm 7.92$ | $-14.12 \pm 7.91$ | $-14.14 \pm 7.92$ | $-21.60 \pm 15.51$ | $-13.2 \pm 7.70$ |
| | | | $w = 0.5$ | | |
| **Interventions** | $17.49 \pm 1.67$ | $7.34 \pm 4.27$ | $17.49 \pm 1.67$ | $13.9 \pm 8.78$ | $0.6 \pm 1.45$ |
| **Returns** | $-14.14 \pm 7.92$ | $-13.9 \pm 7.84$ | $-14.14 \pm 7.92$ | $-15.63 \pm 7.11$ | $-13.01 \pm 7.62$ |
| | | | $w = 1.0$ | | |
| **Interventions** | $17.49 \pm 1.67$ | $4.7 \pm 3.59$ | $17.49 \pm 1.67$ | $15.81 \pm 4.94$ | $0.0 \pm 0.0$ |
| **Returns** | $-14.14 \pm 7.92$ | $-13.69 \pm 7.79$ | $-14.14 \pm 7.92$ | $-13.57 \pm 5.87$ | $-13.23 \pm 7.64$ |
| | | | $w = 2.0$ | | |
| **Interventions** | $17.49 \pm 1.67$ | $2.16 \pm 2.59$ | $17.49 \pm 1.67$ | $8.56 \pm 7.47$ | $0.0 \pm 0.0$ |
| **Returns** | $-14.14 \pm 7.92$ | $-13.38 \pm 7.82$ | $-14.14 \pm 7.92$ | $-18.40 \pm 13.90$ | $-13.4 \pm 7.65$ |

