# OpenReview forum: "Safe Reinforcement Learning using Action Projection: Safeguard the Policy or the Environment?"
_TMLR — Accepted by TMLR_

### Review · Reviewer_pag5 · 2025-12-21

**Summary Of Contributions:**

### 1. Summary

The paper provides a theoretical and empirical comparison between two common approaches for projection-based safe reinforcement learning: Safe Environment RL (SE-RL) and Safe Policy RL (SP-RL). The authors propose a unified framework to formalize these methods and prove that, theoretically, both approaches share the same optimal value functions. A key contribution is the analysis of "action aliasing," which manifests differently in each setting (flat-lining critic in SE-RL vs. rank-deficient Jacobians in SP-RL). To address the limitations of SP-RL, the paper introduces a "Penalty Critic" aimed at mitigating the zero-gradient problem. Experiments on standard control tasks (Pendulum, Quadrotor) and a navigation task are conducted to validate the proposed methods against baselines.

### 2. Strengths

- **Clarification of Theoretical Equivalence:** The paper offers a rigorous proof showing that SE-RL and SP-RL target equivalent optimal value functions. This theoretical grounding clears up ambiguities in the existing literature regarding the fundamental relationship between safeguarding the policy versus the environment.

- **Mechanism Analysis:** The analysis of how "action aliasing" negatively affects learning is insightful. Specifically, distinguishing between the "flat-lining critic" in SE-RL and the "zero-gradient/rank-deficient Jacobian" problem in SP-RL provides a useful taxonomy for understanding failure modes in safe RL.

- **Sound Mitigation Strategy:** The proposed "Penalty Critic" for SP-RL  is mathematically well-motivated and serves as a valid strategy to re-introduce gradient information that is otherwise lost during projection.

### 3. Weaknesses

- **High Computational Cost:** As the authors acknowledge, the improved SP-RL methods incur a significant computational overhead ($2-10 \times$) compared to SE-RL due to the need for differentiating through the optimization layer. This substantial cost somewhat diminishes the practical appeal of the proposed method, especially since the performance gains are not always drastic.

- **Mixed Experimental Results:** The empirical advantage of the proposed SP-RL method is not consistent across all tasks. For instance, in the Seeker environment, the simpler SE-RL baseline (with penalties) performs comparably to or even better than the more complex SP-RL variants. This suggests that the "Penalty Critic" is not universally superior and may depend heavily on the alignment between safety constraints and task objectives.

- **Limited Evaluation Scope:** The experiments are limited to relatively low-dimensional tasks (Pendulum, Quadrotor, Point-mass Seeker). It remains unclear how the "rank-deficient Jacobian" issue and the proposed remedy scale to high-dimensional state/action spaces where action aliasing might be more severe and the computational cost of projection layers would be even more prohibitive.

**Audience:**

Yes

**Audience Explanation:**

Safe Reinforcement Learning is a significant and active area of research within the TMLR community. This paper addresses a fundamental practical question: whether to integrate safety filters into the policy or the environment. The theoretical insights regarding "action aliasing" and the comparative analysis of failure modes (flat-lining critic vs. rank-deficient Jacobians) provide valuable guidance for researchers and practitioners working on constrained optimization and safety-critical control.

**Claims And Evidence:**

Yes

**Claims Explanation:**

The paper's theoretical claims are supported by rigorous mathematical derivations. Specifically, the proof of Theorem 1, which establishes the equivalence of optimal value functions between SE-RL and SP-RL, is sound. The subsequent analysis identifying the distinct mechanisms of "action aliasing" in both frameworks — characterizing them as "flat-lining critics" in SE-RL and "rank-deficient Jacobians" in SP-RL — is logically convincing and well-supported by the formal definitions.

Regarding the empirical claims, the authors provide clear evidence for the effectiveness of their proposed "Penalty Critic" strategy. The experiments on the Pendulum and Quadrotor tasks  convincingly demonstrate that the proposed method helps recover performance in SP-RL settings. Furthermore, the evidence is presented accurately and transparently: the authors candidly report that in the "Seeker" navigation task, the simpler SE-RL baseline remains competitive. This inclusion of mixed results enhances the credibility of the submission, as it avoids overclaiming the universality of the method and instead provides a nuanced view of performance across different task types.

**Requested Changes:**

###

- **Expanded Discussion on Computational Trade-offs (Critical):** While the paper acknowledges a $2-10\times$ computational overhead for improved SP-RL methods due to sensitivity computations, the current discussion could be more prescriptive for practitioners. I request the authors add a paragraph to the Discussion or Conclusion section that explicitly guides readers on when the performance gains of SP-RL justify this cost. Specifically, under what task characteristics (e.g., safety-criticality vs. training speed, state-space complexity) should a practitioner prefer the computationally expensive SP-RL over the robust and cheaper SE-RL baseline?
- **Wall-Clock Time Analysis (Critical):** The paper mentions that detailed wall-clock analysis was excluded due to software-specific variations. However, providing even a rough comparison (e.g., a simple table showing average seconds per training step for one environment) would significantly strengthen the evidence regarding the computational cost mentioned above. This would give readers a concrete sense of the "price" of using differentiable projection layers.
- **Clarity on "Penalty Critic" Implementation (strengthen):** It would be beneficial to slightly expand the description of the "Penalty Critic" implementation details in the main text or appendix to ensure full reproducibility, specifically regarding how the penalty scale hyperparameter ($w$) interacts with the critic's loss function stability during the early phases of training.

---

> ### Author Response · Authors · 2026-02-12
>
> Thank you for your constructive and insightful feedback. We appreciate the opportunity to clarify the practical implications and implementation details of our work. Below are our responses to the points raised:
>
> - **Wall Clock Time Analysis:** As requested, we conducted a wall clock time analysis and added it as a table to appendix A.9. This table provides the average seconds per training step (averaged over 10,000 steps) for all environments which we use for an extended discussion as described next.
>
> - **Discussion on Computational Trade-offs:** To provide better guidance for the readers on the computational trade-offs between performance and computational cost in SP-RL, we have added an additional paragraph to the Discussion section. Essentially, the gap between SE-RL and SP-RL is lowest for environments for which a safe action set is readily available (in our case, the seeker environment) and the projection problem in Eq. (12) reduces to a point containment problem as detailed in appendix A.1. If only a safe state set is available, the constraints in (12b) formalize a set containment problem, which is more complex. Here, computation time increases with state space complexity. For the time being, SP-RL may be computationally prohibitive for environments with safe state sets and a large number of states. However, we expect recent advances in differentiable optimization layers (Grontas et al., 2025; Nguyen et al., 2025) to significantly reduce this gap. Furthermore, it is often possible to decouple high-dimensional safety-critical dynamics into several subproblems which can then be solved in parallel during both the forward and the backward pass. An example of this is provided in Chen et al. (2021) for a distribution network with 37 buses.
>
> - **Improved Evaluation Scope:** To extend the scope of our evaluation, we have introduced a new environment, an energy system optimization task with an observation space featuring 23 dimensions. We observe the same pattern as for the other tasks: Vanilla SP-RL leads to non-convergent runs, but SP-RL with improvements can outperform SE-RL. Due to the theoretical foundation behind the vanishing gradient problem and its remedies, we expect these insights to generalize to even higher-dimensional settings.
>
> - **Clarity on Penalty Critic Implementation:** We added appendix A.7 to provide further details, including an analysis of the evolution of the loss term of the penalty critic for the different penalty factors for all environments. While for the pendulum, the seeker, and the newly added energy management system task, all penalty factors enable convergence of the loss term, it is divergent for low values of $w$ in the quadrotor task. This highlights that task-specific tuning of $w$ is a critical design choice, consistent with findings in Bejarano et al. (2024) for SE-RL.
>
> **References**
>
> Bejarano, Federico Pizarro, Lukas Brunke, and Angela P. Schoellig. "Safety filtering while training: Improving the performance and sample efficiency of reinforcement learning agents." IEEE Robotics and Automation Letters, 10(1):788–795 (2025).
>
> Chen, Bingqing, et al. "Enforcing policy feasibility constraints through differentiable projection for energy optimization." Proceedings of the Twelfth ACM International Conference on Future Energy Systems. 2021.
>
> Grontas, Panagiotis D., et al. "Pinet: Optimizing hard-constrained neural networks with orthogonal projection layers." arXiv preprint arXiv:2508.10480 (2025).
>
> Nguyen, Hoang T., and Priya L. Donti. "FSNet: Feasibility-seeking neural network for constrained optimization with guarantees." arXiv preprint arXiv:2506.00362 (2025).

---

### Review · Reviewer_t7Dp · 2026-01-30

**Summary Of Contributions:**

**Summary**

This paper conducts a systematic comparison between two projection-based safety-enforcing mechanisms in reinforcement learning, safe environment (SE) and safe policy (SP). The paper presents theoretical analyses that establish the equivalence and difference of the two approaches under certain settings. It also demonstrates the effect of action aliasing across both approaches and proposes a variant, PenC, to improve SP. Some experiments on RL benchmarks validate the theoretical claims.

**Audience:**

Yes

**Audience Explanation:**

**Weaknesses**

I have two minor comments on the presentation and experiments.

1. Limited theoretical explanations of action aliasing

   The paper theoretically explains the connection between SE and SP in section 6 and presents the solutions to action aliasing. But I find that the formal definition and explanation of action aliasing itself is lacking and somewhat hidden in the discussion of section 6.3 on deterministic policies. Besides, I also feel that the motivation behind the newly proposed PenC approach should be more carefully explained.

2. Scope of experiments

   The experiments compare the performance of SE and SP in terms of return and safety intervention, which are two primary metrics for the task. I believe it would also be useful to include experiments on the practical computational efficiency of the two approaches (as mentioned in the discussion section), since this would provide guidance for practitioners when choosing between them.

**Questions and Minor issues**

1. Page 2

   What does "the sensitivity of the safe action with respect to the unsafe action" mean?

2. Page 5

   $\mathbf{g}_t$ => $g_t$.

3. Page 6

   The order of $u, x$ in $p_x^{SE}(x' | u, x)$ and $p_x^{SP}(x' | u, x)$ seems inconsistent with THE definition of $p_x$.

4. Page 7

   $\Psi_{\pi^\perp}^{SP}$ is not clearly explained here.

5. Page 7 and 11

   Border and boundary are used interchangeably.

6. Page 9

   Please define normal cone formally.

7. Page 10, 11

   Additional fullstop after the equations for $q_\pi^{aug}$ and the figure on page 11.

8. Page 11

   $\mu$ seems not defined.

9. Page 13

   Figure 5 seems to be mentioned before Figure 4.

**Broader Impact Concerns:**

N.A.

**Claims And Evidence:**

Yes

**Claims Explanation:**

**Strength**

I'm not an expert in RL, but I find this paper clearly motivated and well written. I also think a systematic comparison between two existing paradigms is important.

**Requested Changes:**

See weaknesses and questions above.

---

> ### Author Response · Authors · 2026-02-12
>
> We are very grateful for your time and your effort in helping us improve this paper. Below we summarize the changes that we made based on your input:
> - **Improved Theoretical Explanation of Action Aliasing:** We have added a more explicit formal definition of action aliasing in section 6.3 (lemma 2).
> - **Motivation behind Penalty Critic:** We have added another advantage of the penalty critic over the per-sample loss to section 7.2, which is that the per-sample loss can only push unsafe actions to the boundary of the safe set, but not to its interior. We hope that this improves the motivation as requested.
> - **Improved Scope of Experiments:** To provide better guidance for practitioners wanting to choose between SE-RL and SP-RL, we have added a wall clock time analysis in appendix A.9. It compares the average seconds per training step for TD3 in all environments. We discuss the results in section 9 as summarized in our answer to Reviewer pag5.
> - **Question 1:**  With the term "sensitivity", we are referring to how changes in the unsafe action selected by the RL agent affect the safe action that is the solution to the optimization problem in Eq. (12). This is also defined mathematically in section 5.2.1.
> - **Question 6:** We have included a definition of the normal cone of a point on the boundary of a convex set in lemma 2 in section 6.3.
> - **Question 8:** The mean $\mu$ is defined in the footnote on page 8 of the paper. To improve readability, we have also added a brief explanation at the position in the main text that you indicated (section 7.2), such that it now says "the mean $\mu$ of the Gaussian policy $\mathcal{N}(\mu, \Sigma)$".
> - **Other Questions:** Thank you for finding these inconsistencies, which we have addressed through the respective changes in the new version of the paper.

---

> > ### Comment · Reviewer_t7Dp · 2026-02-15
> >
> > Thank you for the response. I do not have other comments. Please double-check the references to ensure they are up to date. Some of the arxiv papers are already published. e.g.,
> >
> > 1. Pinet: Optimizing hard-constrained neural networks with orthogonal projection layers
> > 2. FSNet: Feasibility-Seeking Neural Network for Constrained Optimization with Guarantees

---

> > > ### Author Response · Authors · 2026-02-16
> > >
> > > Thank you for pointing this out, we will double-check the references.

---

### Review · Reviewer_oNPV · 2026-01-31

**Summary Of Contributions:**

This paper considers differences between two approaches to safe RL: one where a safeguard mapping unsafe to safe actions is put into the environement (SE-RL), and one where this safeguard is put into the policy (SP-RL) such that policy gradients are calculated through the safeguard.

The paper makes a number of theoretical contributions: it shows that in some ways SE-RL and SP-RL are equivalent approaches. Value functions optimal for SE-RL are also optimal for SP-RL. Further, for stochastic policies, both have identical policy and critic gradients, meaning that both will converge to the same local optima. For deterministic policies, it is shown that both SE-RL and SP-RL suffer from action aliasing, where several unsafe actions are mapped to the same safe action, but that both are affected differently by it.

The paper then considers strategies for mitigating the effects of aliasing on both SE-RL and SP-RL. In SE-RL settings, prior work has commonly introduced a penalty into the reward function that penalizes the difference between an unsafe action and its safe replacement. This penalty therefore has a non-myopic effect on the optimal policy. For SP-RL, prior work has proposed to introduce additional terms into the policy loss, also penalizing distances between unsafe and corresponding safe actions, but doing so in a myopic way. This paper therefore proposes for SP-RL to introduce an additional penalty critic. This yields policy gradients that push unsafe actions closer to safe actions, and analogues to the SE-RL approach, but does so in a non-myopic manner.

Experimentally the paper shows that the proposed penalty critic for SP-RL improves return and safety for SP-RL, and outperforms the myopic loss penalty approaches from prior work on two out of three tasks considered.

**Audience:**

Yes

**Audience Explanation:**

This is a relevant paper or people engaged in Safe-RL research. The paper does an fantastic job presenting safeguards in Safe RL and provides an excellent overview of related work, making it a valuable introduction to the topic for a broader audience of RL/ML researchers.

**Broader Impact Concerns:**

No concerns.

**Claims And Evidence:**

Yes

**Claims Explanation:**

Some of the core results in the paper are theoretical in nature and have accompanying proofs in the appendix. The proposed penalty critic for SP-RL has theoretical justification and is tested empirically. Empirical results could have been more extensive, with experiments carried out in only three relatively simple environments, but are sufficient in combination with the theoretical contributions.

**Requested Changes:**

I have no issues that would prevent me from recommending acceptance.

I do have some minor points:
1. In section 5.2 when introducing the SP-RL setting, you explain that action adjustment is part of the policy. Yet when describing the MDP you introduce new transition and reward densities $p_x^{SP}$ and $p_r^{SP}$, despite these being exactly the same as in the original MDP. For SE-RL it made sense to introduce new densities as these now included the safeguard, but would it not be better to simple use the original $p_x$ and $p_r$ for SP-RL to emphasize the fact that the MDP does not change?
2. On page 11, first paragraph, you write “policy actions that are inherently safe or closer to the feasible region are preferred” Is it not supposed to be “safe region” rather than “feasible region”?

---

> ### Author Response · Authors · 2026-02-12
>
> We thank the reviewer for their time and the valuable remarks. We have corrected the wording on page 11 to "safe region". Regarding the other comment, our motivation for introducing $p^\text{SP}_x$ and $p^\text{SP}_r$ was that the support of the densities changes compared to the unsafe MDP with $p_x$ and $p_r$. We have made this distinction explicit in the text in section 5: "Please note that the support of these densities changes compared to the original MDP introduced in section 3.1". We hope that this clarifies why we chose this notation, and we are looking forward to any further questions you might have.
>
> Furthermore, we have introduced an additional environment, an energy management system task with 23 observations. Similar to the other environments, we observe convergence issues with vanilla SP-RL, which are mitigated most effectively using our proposed penalty critic.

---

### Decision · Action_Editor_1Bd2 · 2026-03-10

**Recommendation:** Accept as is

**Audience:**

Yes

**Audience Explanation:**

Yes, this paper would be of interest to researchers interested in safe reinforcement learning and control.

**Claims And Evidence:**

Yes

**Claims Explanation:**

The paper studies a very common technique in safe reinforcement learning: when a policy proposes an unsafe action, a safety filter automatically replaces it with the closest safe action before executing it. This approach is widely used in robotics, control, and safety-critical systems because it ensures that unsafe actions are never executed. The central question of the paper is: where should this safety filter be placed in the learning architecture? There are two main options: (a) the safety filter is treated as part of the environment; the policy outputs an action as usual; if the action is unsafe, the environment automatically changes it to a safe one, and (b) the safety filter is placed inside the policy itself; when the policy produces an action, the safety filter modifies it if necessary and the learning algorithm explicitly accounts for this modification during training. The advantage of the latter is that the agent explicitly learns how safety constraints affect actions.

Reviewers found that their questions and concerns were addressed well and recommended that the paper be accepted.